**METHOD**

# spVC for the detection and interpretation of spatial gene expression variation

Shan Yu[1*†] and Wei Vivian Li[2*†]

†Shan Yu and Wei Vivian Li contributed equally to this work.

*Correspondence:
sy5jx@virginia.edu; weil@ucr.edu

[1] Department of Statistics, Unversity of Virginia, Charlottesville 22903, VA, USA
[2] Department of Statistics, University of California, Riverside 92521, CA, USA

**Abstract**

Spatially resolved transcriptomics technologies have opened new avenues for understanding gene expression heterogeneity in spatial contexts. However, existing methods for identifying spatially variable genes often focus solely on statistical significance, limiting their ability to capture continuous expression patterns and integrate spot-level covariates. To address these challenges, we introduce spVC, a statistical method based on a generalized Poisson model. spVC seamlessly integrates constant and spatially varying effects of covariates, facilitating comprehensive exploration of gene expression variability and enhancing interpretability. Simulation and real data applications confirm spVC's accuracy in these tasks, highlighting its versatility in spatial transcriptomics analysis.

## Background

Advances in spatially resolved transcriptomics technologies have enabled transcriptome profiling in single cells while retaining the spatial information, providing new opportunities to understand gene expression heterogeneity in the spatial contexts. A critical task in the analysis of spatial transcriptomics data is to discover spatially variable genes (SVGs) that have distinct expression patterns across different spatial locations [1]. Spatial variation of gene expression can reflect the state of cells in specific locations, ligand-receptor interactions, and cell-cell communications [2, 3]. Therefore, SVGs are also important for understanding disease microenvironment and identifying potential therapeutic targets [4]. For example, Wang et al. [5] identified spatially variable metabolic genes in tumor tissues of prostate cancer patients and suggested further investigation of spatial metabolic heterogeneity to guide targeted therapy; Chen et al. [6] identified genes whose expression levels increased gradually with accumulating amyloid plaques in spatial transcriptomics data for studying Alzheimer's disease. In addition to their capacity for yielding valuable biological insights, SVGs also play significant roles in the dimensionality reduction of spatial data, which is a necessary step in standard spatial transcriptomics analysis [2, 7].

Diverse computational methods have been developed for the identification of SVGs from spatial transcriptomics data [2, 8–13]. Since observed expression levels of a gene may present spatial variation due to random noises, the identification of SVGs is typically formulated as a statistical problem to determine if the observed spatial variation is statistically significant. For example, SpatialDE [14] uses normalized expression levels of a gene to construct a Gaussian process regression model, where the spatial locations are incorporated through a spatial covariance function. The statistical significance of spatial variation is evaluated by a likelihood ratio test between models with and without the spatial covariance. SPARK [15] uses a generalized Poisson regression model to directly fit the read counts, and also uses the Gaussian process to model spatial correlation patterns. Then, SPARK evaluates the statistical significance using a score test. The SPARK-X method [16], on the other hand, uses a non-parametric test to evaluate the similarity between the gene expression covariance matrix and the spatial distance covariance matrix. The MERINGUE method [17] uses the spatial autocorrelation measure, Moran's *I*, and its statistical significance to quantify the association between gene expression and spatial locations.

Most existing methods focus on the identification of SVGs based on their statistical significance, and are subject to two major limitations. First, it is difficult to quantify how gene expression changes in a spatial domain with existing models. The continuous expected expression patterns across the spatial domain cannot be easily estimated by the generalized regression models with Gaussian processes or non-parametric methods. Second, most existing methods cannot incorporate cell-level (or spot-level) covariates in the modeling process, making it challenging to comprehensively analyze the dependence between gene expression and distinct types of cellular characteristics, such as spatial locations and cell types or states. Another drawback of methods unable to account for cell/spot covariates lies in their incapability to distinguish between cell-type-specific genes, which exhibit spatial variation due to the non-random distribution of different cell types, and genes that, while not inherently cell-type-specific, display spatial variation resulting from local cellular communications or other underlying factors.

To address the above challenges in identifying and interpreting SVGs from spatial transcriptomics data, we propose a statistical method named spVC, which is based on a generalized Poisson model that allows *sp*atially *V*arying *C*oefficients of cell/spot-level covariates. We would like to summarize three key novel features of spVC. First, spVC integrates constant and spatially varying effects of cell/spot-level covariates, enabling a comprehensive exploration of how spatial locations and other covariates collectively contribute to gene expression variability. By incorporating various sources of information such as cell types, cell states, or regulation factor activities, spVC serves as a versatile tool for investigating diverse biological questions. Second, spVC offers statistical inference tools for each of the constant or spatially varying coefficient, providing a statistically principled approach to selecting different types of SVGs. Third, in addition to statistical significance, spVC is able to estimate the expected effect of spatial locations and other covariates on gene expression in the designated spatial domain. This additional layer of information facilitates the characterization and interpretation of identified SVGs, enhancing our ability to understand their functional implications. In summary,

we anticipate that the spVC method will improve our capability to detect, interpret, and comprehend variation of gene expression with spatial transcriptomics data.

## Results

### An overview of the spVC method

The spVC method aims to provide a flexible spatial regression model and corresponding inference tool to elucidate the relationships between gene expression, spatial locations, and other cell/spot-level covariates in a streamlined approach. To highlight the importance of incoporating covariate information in the interpretation of SVGs, we consider three hypothetical gene examples corresponding to three different scenarios (Fig. 1A). We assume the studied spatial domain is a square region, and there is one spot-level covariate which represents the proportion of a specific cell type at the spots (Fig. 1B). In scenario 1, the gene's true expression only depends on a constant (i.e., spatially invariant) effect of the cell type proportion, and there are no other factors that would introduce additional spatial patterns. Since the cell type proportion changes in the spatial domain, the observed read counts would present spatial variation (Fig. 1C). In scenario 2, the gene's expression depends on spatially varying effects of the cell type proportion (Fig. 1D), and there are no other factors

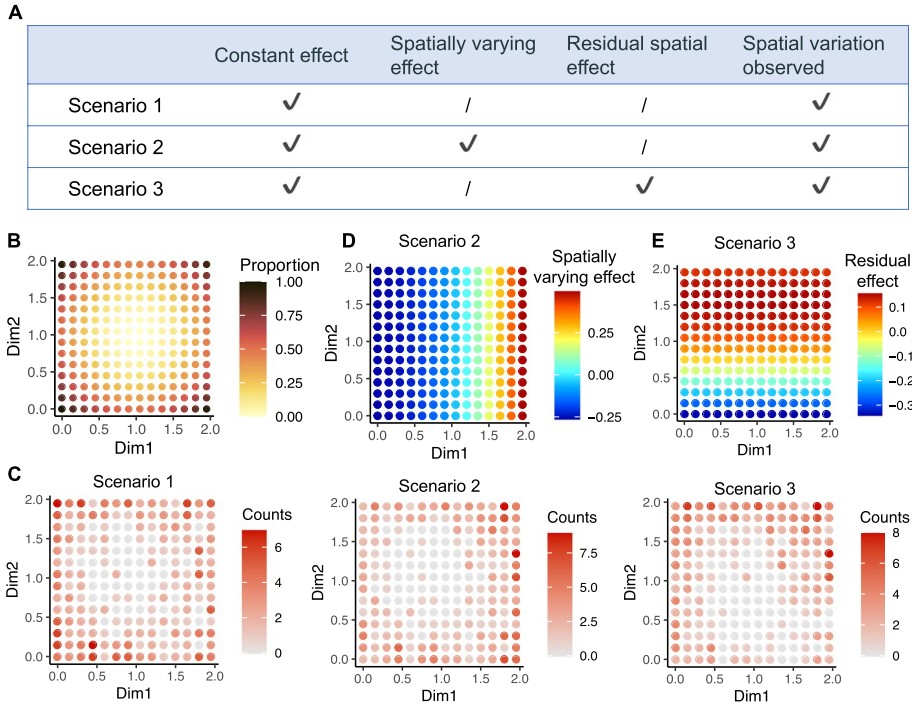

**Fig. 1** Spatial variation of gene expression introduced by different factors. **A** Three hypothetical genes with observed spatial variation introduced by different underlying factors. "Constant effect" and "Spatially varying effect" refer to constant and spatially varying coefficients of the covariate, respectively. "Residual spatial effect" refers to spatial effects independent of the covariate. A "✓" means that the effect is present. **B** Spot-level cell type proportions. **C** Read counts generated from Poisson distributions, whose mean parameters are based on the scenarios specified in **A**. For simplicity, we assume no difference in library size across spots. **D** Spatially varying effects of the cell type proportion in scenario 2. The average effect is 0 in this square domain so it remains identifiable. **E** Residual spatial effects in scenario 3. The average effect is 0 in this square domain so it remains identifiable from the constant effect of cell type proportion

that would introduce additional spatial patterns. In scenario 3, the gene's expression depends on a constant effect of the cell type proportion, and there exists a residual spatial effect that is independent of the cell type proportion (Fig. 1E). Even though the three example genes all present spatial variation in their observed counts (Fig. 1C), the underlying factors that regulate gene expression are apparently different. In real data applications, only the read counts are observed, so statistical tools are in need to disentangle the relationships between gene expression, spatial locations, and other available covariates.

The spVC method requires spatial transcriptomics data (the read count matrix and the corresponding spatial location matrix) and spot-level covariate data as its input (Fig. 2A). For every gene, spVC constructs a generalized Poisson regression model with the response variable being the read counts, and the explanatory variables being the spatial coordinates and cell/spot-level covariates (Methods; Fig. 2B). For example, the covariates could be cell type labels (categorical), cell type proportions (continuous), or activities of relevant regulation factors (continuous or binary). To improve the interpretation of spatial expression variation, spVC decomposes the expected gene expression in every cell/spot into four components: the intercept, the constant (spatially invariant) effect of the covariates, the spatially varying effects of the covariates, and the unexplained residual spatial effects. The estimation of the spVC model is achieved using a Quasi-Poisson approach to account for potential over-dispersion in spatial transcriptomics data. For the estimation of spatial effects, spVC uses multivariate splines and is flexible to learn diverse spatial patterns. For each constant or spatial effect, spVC also evaluates its statistical significance and outputs the corresponding *P* value (Fig. 2C-D).

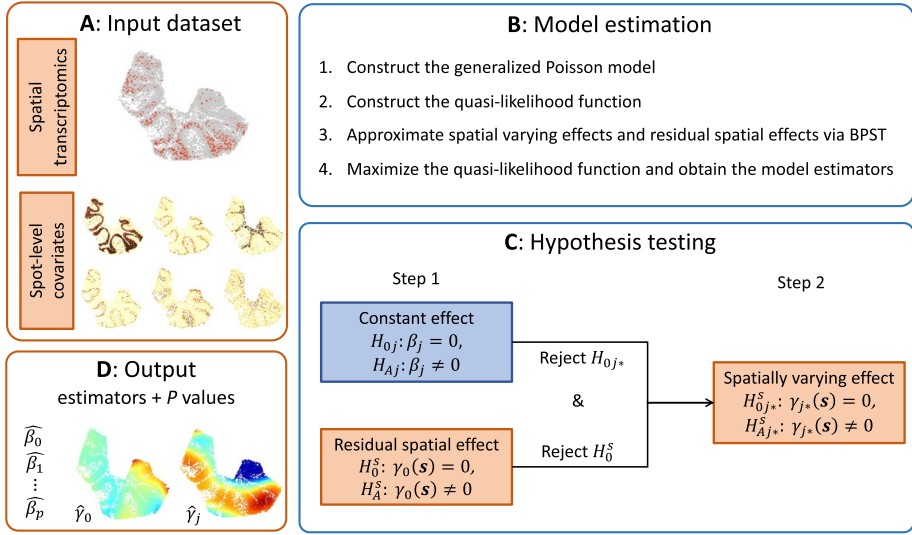

**Fig. 2** Overview of the spVC method. **A** The required input of spVC includes the spatial transcriptomics data (the read count matrix and the corresponding spatial location matrix) and the spot-level covariate data. The covariates should be provided for the same spots observed in the spatial transcriptomics data. **B** The four main steps in spVC's estimation procedure. **C** The two-step testing procedure used in this article. **D** For each gene, spVC outputs the estimated constant and spatial effects as well as their corresponding *P* values

### spVC accurately detects spatial gene expression patterns given spot-level covariates

The observed gene expression levels in real spatial transcriptomics data reflect the combined effects of cell/spot-level covariates and any residual spatial patterns unexplained by available covariates, but the ground truth information remains unknown. Therefore, we first designed a simulation study to generate synthetic spatial transcriptomics data with known underlying expression patterns (see the "Methods" section for details). Using our simulation procedure, we generated five datasets with 20,000 genes and varying spot numbers (500, 1000, 2000, 5000, and 8000). The corresponding spatial coordinates and spot covariates (proportions of four cell types) were directly sampled from a real spatial transcriptomics dataset of mouse cerebellum [18] (Fig. 3A). In every simulated dataset, we considered four gene groups, each with 5000 genes. In Group 1, the expected gene expression did not depend on cell type proportions or spatial coordinates (no covariate effect + no residual spatial effect). In Group 2, the expected gene expression depended on cell type proportions but not on spatial coordinates (constant covariate effect + no residual spatial effect). In Group 3, the expected gene expression depended on both cell type proportions and spatial coordinates (constant covariate effect + residual spatial effect). In Group 4, in addition to the assumption on Group 3, the coefficients of two cell type proportions were spatially varying (spatially varying covariate effect + residual spatial effect).

We applied spVC and another four methods developed for SVG detection, SpatialDE [14], SPARK [15], SPARK-X [16], and MERINGUE [17], to the simulated data. SpatialDE and MERINGUE cannot directly account for covariate information in their SVG detection models, so the covariates were regressed out from normalized gene expression before SVG detection (Methods). We first calculated their type I errors and statistical power for detecting the residual spatial effects, after applying a threshold of 0.05 to $P$ values adjusted by the Benjamini-Hochberg (BH) procedure [19]. We found that spVC,

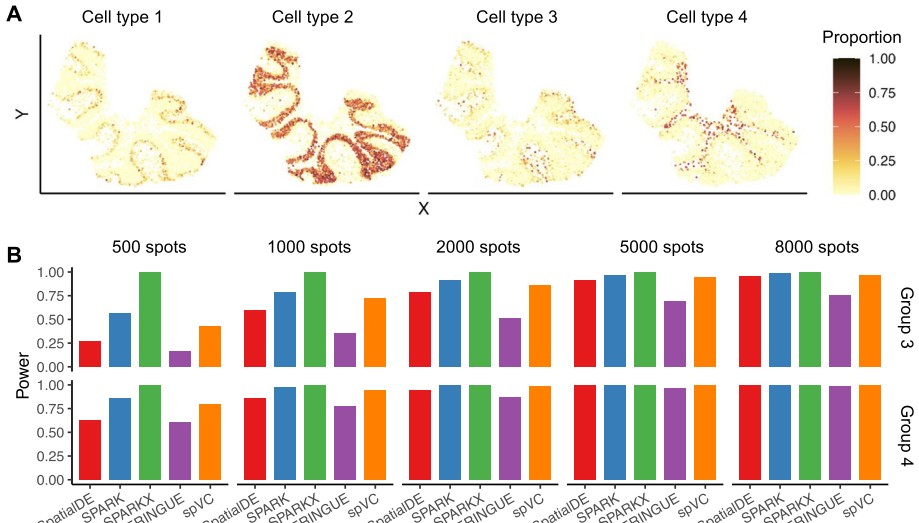

**Fig. 3** Comparison of spVC and four alternatives methods on simulated data. **A** Proportions of the four cell types in the simulated spatial transcriptomics data with 5000 spots. **B** Statistical power of the five methods on genes in Groups 3 and 4 for detecting residual spatial gene expression variation in the presence of covariates

SpatialDE, and MERINGUE could control the type I errors below 0.05 for genes in both Group 1 and Group 2 (Additional file 1: Fig. S1), indicating that they are able to distinguish between the covariate effects and unexplained residual spatial effects. SPARK achieved a good control of type I error on Group 2 when the spot number was relatively small, but had an error of 0.063 and 0.241 when spot number was 5000 and 8000, respectively. SPARK-X would identify most genes in Groups 1 and 2 as SVGs after accounting for the cell type proportion information, even though the simulated gene expression did not directly depend on spatial coordinates. As for the statistical power, spVC consistently demonstrated higher power than SpatialDE and MERINGUE, and both spVC and SpatialDE had a power $> 0.9$ when the spot number was a least 5000 (Fig. 3B).

Next, using genes in Group 4, we evaluated spVC's power in detecting the spatially varying effects of cell type proportions. The other four methods were not designed for this goal and, as a result, were not included in this analysis. In our simulation, the proportions of cell type 2 and cell type 4 had spatially varying effects on the expected gene expression in Group 4, and spVC had a power $> 0.785$ for both cell types when the spot number was at least 5000 (Fig. 4A). We also noticed a consistent trend where the power for cell type 2 remained higher than that for cell type 4. This observation can likely be attributed to the higher abundance of the former in the simulated spatial data.

An important feature of spVC is its ability to simultaneously estimate covariate effects and residual spatial effects on gene expression in addition to evaluating the statistical significance of spatial variation. To demonstrate the estimation ability of spVC, we selected one example gene from each of the four groups (from the dataset with 5000 spots). Gene 1 was from Group 1 without any covariate or spatial effects; the expression of Gene 2 (from Group 2) depended on the proportions of cell types 1 and 2 (with constant effects); the expression of Gene 3 (from Group 3) depended on the proportions of cell types 3 and 4 (with constant effects) and residual spatial effects; the expression of Gene 4 (from Group 4) depended on the proportions of cell types 2 and 4 (with spatially varying effects) and residual spatial effects (Fig. 4B). It was not possible to distinguish these different effects by visualizing the normalized gene expression data (Fig. 4C). However, the residual spatial effects estimated by spVC correctly reflected the fact that only the expression of Gene 3 and Gene 4 directly depended on spatial coordinates (Fig. 4D–E). The $P$ values of these four genes were $9.533 \times 10^{-1}$, $1.000$, $1.268 \times 10^{-7}$ and $2.220 \times 10^{-16}$, respectively. In addition, spVC was able to estimate the spatially varying coefficients of cell type proportions for Gene 4, which were in close agreement with the true coefficients (Fig. 4F).

Besides the above individual gene examples, we compared the complete sets of true and estimated values for different components. This encompassed intercepts (for all 4 groups; Additional file 1: Fig. S2), constant effects of the covariates (for Groups 2 to 4; Additional file 1: Fig. S3), spatially varying effects of the covariates (for Group 4; Additional file 1: Fig. S4), and unexplained residual spatial effects (for Groups 3 and 4; Additional file 1: Fig. S5). The results provided further validation of spVC's capacity to accurately discern the directions of association between gene expression and covariates, as well as its precision in estimating covariates' and residual spatial effects across genes. Notably, in Group 4, as spot size increased from 500 to 8000, the median Pearson correlation between true and estimated residual spatial effects (evaluated at the

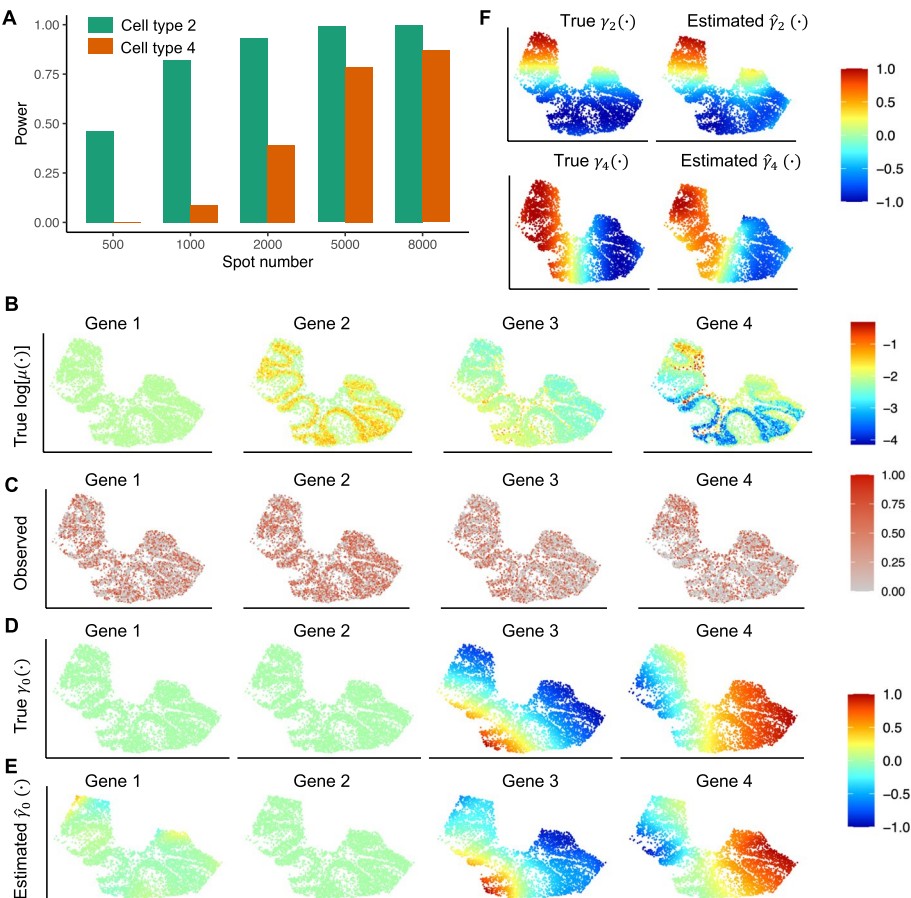

**Fig. 4** spVC's estimation and inference on simulated data. **A** spVC's power in detecting spatially varying covariate effects. **B** True log-transformed expected expression, $\mu(\boldsymbol{x}_i, \boldsymbol{s}_i)$ ($i = 1, \ldots, 5000$), of the four example genes. **C** Relative expression levels of the four example genes based on the simulated data. The simulated counts were normalized by library size, log-transformed, and then scaled by the min-max normalization to obtain the relative expression levels. **D** True spatial effects, $\gamma_0(\boldsymbol{s}_i)$ ($i = 1, \ldots, 5000$), of the four example genes. Data were scaled to the range of $[-1, 1]$ for visualization. **E** spVC's Estimated spatial effects, $\hat{\gamma}_0(\boldsymbol{s}_i)$ ($i = 1, \ldots, 5000$), of the four example genes. Data were scaled to the range of $[-1, 1]$ for visualization. **F** (Top) True and estimated spatially varying effects of cell type 2's proportion, $\gamma_2(\boldsymbol{s}_i)$ and $\hat{\gamma}_2(\boldsymbol{s}_i)$ ($i = 1, \ldots, 5000$), of Gene 4. (Bottom) True and estimated spatially varying effects of cell type 4's proportion, $\gamma_4(\boldsymbol{s}_i)$ and $\hat{\gamma}_4(\boldsymbol{s}_i)$ ($i = 1, \ldots, 5000$), of Gene 4. Data were scaled to the range of $[-1, 1]$ for visualization

observed spots) rose from 0.75 to 0.95. Similarly, the median correlation between true and estimated covariates' spatial effects increased from 0.89 to 0.96. These findings underscore the importance of sample size in spatial variation analysis of sparse transcriptomics data.

In addition to estimation and inference accuracy, we also summarized the computational cost on the simulated datasets (Additional file 1: Fig. S6). The fastest method was SPARK-X, which uses a non-parametric approach. It finished running on all four datasets within 1 minute. spVC was the second fastest, and finished running on all four datasets within 1 h. On the largest dataset with 8000 spots, SpatialDE, MERINGUE, and SPARK took 3.10, 16.60, and 27.25 h to complete, respectively. In terms of maximum memory usage, while spVC initially consumed more memory than

the other methods for smaller datasets, its memory requirement increased at a slower rate as the number of spots reached 8000.

Lastly, we performed a simulation study using an independent simulator named SRTsim [20], which allowed us to generate spatial transcriptomics data while specifying whether a gene had constant covariate effects. We used SRTsim to generate six simulated datasets (see the "Methods" section for details). Each dataset included 150 genes with constant covariate effects (similar to Group 2 genes in our previous simulation) and 850 genes with no spatial or covariate effects (similar to Group 1 genes in our previous simulation). Then, we evaluated the type I errors of different methods in identifying residual spatial effects. Consistent with our previous simulation findings, we observed that all methods except for SPARK-X had a type I error of 0; SPARK-X's type I error was between 0.86 and 0.90. Additionally, we noted that most methods demonstrated improved type I error control in this simulation, likely attributed to the constraint of setting a constant mean and dispersion parameter (of the Negative Binomial distribution) for all genes, thereby reducing the complexity of the simulated data.

### Application of spVC to human cortex data

For the first real data application, we applied spVC to spatial transcriptomics data of the six-layered human dorsolateral prefrontal cortex sequenced by the 10x Genomics Visium platform [21]. After filtering genes that were detected in fewer than 100 spots, this dataset included 11,242 genes and 3611 spatial spots. The human cerebral cortex has a laminar organization, in which different layers demonstrate layer-specific gene expression patterns [21]. Therefore, it would be useful to distinguish spatial variation of gene expression led by the laminar organization from the spatial variation that is independent of layer-specific expression. Spots in the above dataset were annotated by seven layers (six neocortical layers, L1 to L6, and the white matter) (Fig. 5A). We treated these categorical labels as the covariates in the spVC model, with the white matter (WM) as the baseline category. As outlined in the "Methods" section, this dataset fits into the scenario where the categorical covariates exhibit clustered patterns, rather than dispersed patterns. Therefore, we applied the spVC method without the spatially varying covariate effects (i.e., formula (4)) to ensure proper identifiability.

To evaluate the false discovery rate of spVC, we first performed a permutation analysis by randomly permuting the spatial coordinates of the spots. In this permuted dataset, none of the genes were expected to have any significant spatial effects in their expression levels, so we evaluated the type I errors of different methods in identifying residual spatial effects in the presence of covariate information. Using 0.05 as a threshold on adjusted $P$ values, we found that spVC had a type I error of 0.001 on this dataset. For comparison, the type I error rates of SpatialDE, SPARK, SPARK-X, and MERINGUE were 0.000, 0.161, 0.898, and 0.001, respectively (Fig. 5B). To further evaluate the theoretical null distributions of the test statistics ($T_j^{\alpha}$; see the "Methods" section) used in spVC, we repeated the permutation procedure 100 times and obtained 100 $P$ values for each gene. The quantile-quantile plots of the $P$ values for two example genes are shown in Additional file 1: Fig. S7A–B. To quantitatively evaluate the results across all genes, we calculated the root of mean squared error (RMSE) between the observed quantile-quantile curve and the $y = x$ reference line for each gene. The average RMSE was 0.026,

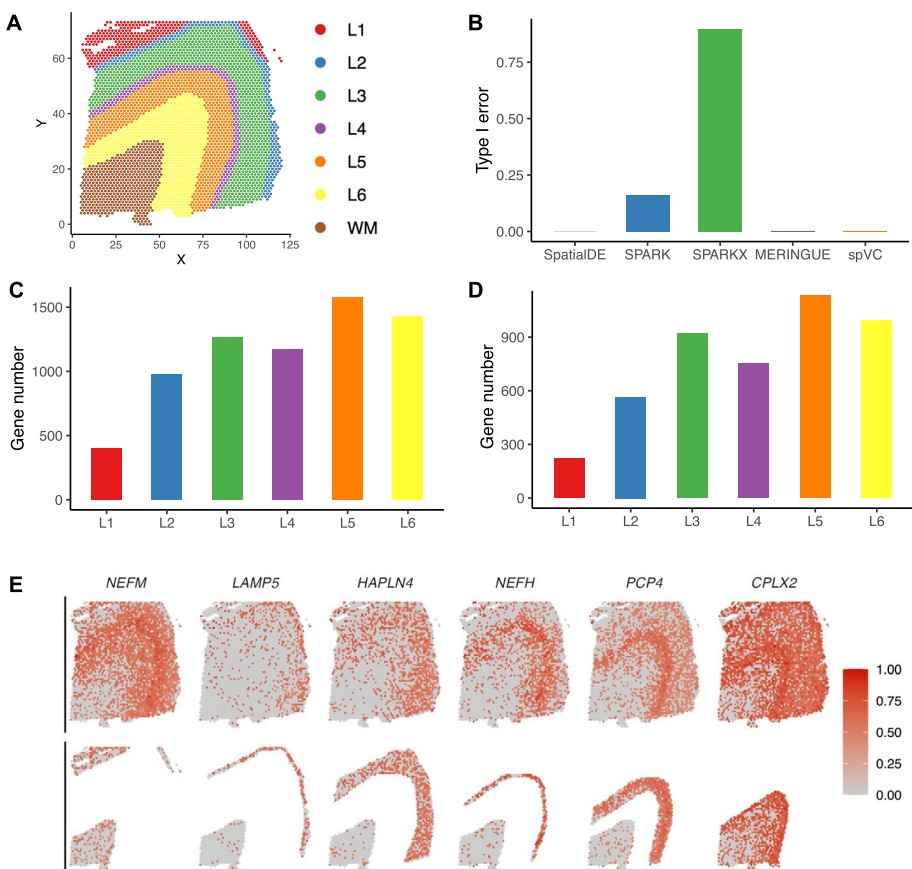

**Fig. 5** Analysis of the prefrontal cortex spatial transcriptomics data. **A** Spatial coordinates and layer annotations of the cortex data. **B** Type I errors of SpatialDE, SPARK, SPARK-X, MERINGUE, and spVC in the permutation analysis. **C** Number of significant genes whose expression was up-regulated in each of the six neocortical layers compared with the white matter. **D** Number of significant genes whose expression was down-regulated in each of the six neocortical layers compared with the white matter. **E** Relative expression levels of six example genes with significantly higher expression in L1 to L6 layers, respectively. The top row shows all spots and the bottom row only shows spots in the corresponding layers and WM . The read counts were normalized by library size, log-transformed, and then scaled by the min-max normalization to obtain the relative expression levels

indicating that the realized null distributions could be well approximated by the theoretical distributions.

We then applied spVC to the real cortex spatial transcriptomics data. Using 0.05 as a threshold on adjusted *P* values of the constant covariate coefficients, we discovered 3593 genes whose expression levels were layer-associated and had a significant difference between WM and at least one of the neocortical layers (Fig. 5C–D). For example, Fig. 5E shows six genes, *NEFM, LAMP5, HAPLN4, NEFH, PCP4,* and *CPLX2*, which had significantly higher expression in L1 to L6 layers, respectively. In addition to the individual gene examples, we also observed layer-specific gene expression patterns among the top genes with the largest regression coefficients of the layer covariates (Additional file 1: Fig. S8).

To further investigate the layer-associated genes identified by spVC, we compared these genes with the cortex layer-associated genes studied in Maynard et al. [21],

which used a model-based method to identify the layers associated with 126 previously reported marker genes. Among these, 92 genes were detected in the spatial transcriptomics data. Maynard et al. reported associations with layers L1 to L6 for 4, 44, 46, 19, 47, and 48 genes, respectively. Our analysis revealed that the percentages of these genes positively associated with the layers, as identified by spVC (using WM as the baseline), were 25.0% (1), 31.8% (14), 41.3% (19), 36.8% (7), 42.6% (20), and 41.7% (20). These genes indeed demonstrated layer-associated expression patterns, and in particular, differential expression between the associated layers and the WM layer (Additional file 1: Fig. S9). It is noteworthy that marker genes are expected to be a subset of significant genes identified by spVC, as spVC can potentially identify both layer-specific and layer-associated genes. We then evaluated the 2097 genes only reported to be layer-associated by spVC, and also observed apparent differences between the identified layers and WM (Additional file 1: Fig. S10). Notably, only two of the six example genes (*NEFH* and *PCP4*) in Fig. 5E were in the previously reported gene list. In contrast, the 67 genes solely reported to be layer-associated in Maynard et al. did not present obvious differences between the neocortical layers and WM (Additional file 1: Fig. S11). These comparisons suggest spVC's ability to identify known and novel tissue-layer-associated genes from spatial transcriptomics data.

Since the cortex has a laminar organization and many genes in the cortex have layer-specific or layer-associated expression, a large portion of these genes would be considered to possess a spatial effect in their expression if the layer-specific information is not taken into account during the modeling process. Instead, we used spVC to investigate which genes presented a significant residual spatial effect after accounting for the layer covariates. Using 0.05 as a threshold on the adjusted *P* values, we discovered 823 genes with a significant spatial effect in their expression after adjusting for the layer covariates. Among these genes, 552 genes overlapped with the 3593 genes whose expression were layer-associated. In comparison, the number of genes with significant residual spatial effects identified by SpatialDE, SPARK, SPARK-X, and MERINGUE were 404, 5068, 11,055, and 496, respectively. We compared these genes and found that the 823 genes discovered by spVC were also identified by at least one other method, with 97.0% of its genes being identified by at least three methods. In contrast, only 19.0% of SPARK genes, 8.7% of SPARKX genes, and 85.1% of MERINGUE genes were identified by at least three methods. We then investigated the 3022 genes which only had constant layer effects in spVC's results but were identified to have significant residual spatial effects by non-spVC methods, and found that they had much smaller estimated residual spatial effects compared with the 823 genes identified by spVC (Additional file 1: Fig. S12). In addition, we evaluated the nine genes that were identified as significant by all methods except for spVC, and they did not present obvious spatial variation that could not be explained by the layer distributions (Additional file 1: Fig. S13).

A unique feature of spVC is that it's not only able to detect the significance of spatial effects in the presence of cell covariates, but also to quantify the expected strength of spatial effects across locations. Therefore, we focused on the 645 genes that had significant residual spatial effects and were detected in more than 500 spots, and performed a hierarchical clustering analysis of these genes based on their estimated residual spatial effects, $\hat{\gamma}_0(\cdot)$ (Methods). We discovered 11 gene clusters with distinct

spatial patterns (Fig. 6A). These discovered patterns were different from those presented by the organization of cortical layers. It is worth noting that, as the observed gene expression levels were confounded by layer covariates (Additional file 1: Fig. S14), such spatial patterns in Fig. 6A can only be revealed by spVC's estimation but not from the observed spatial transcriptomics data.

To further understand the potential biological functions of the above significant genes, we performed a Gene Ontology (GO) enrichment analysis on three sets of genes: 3041 genes whose expression was layer-associated but did not have residual spatial effects ("layer-associated"), 271 genes whose expression had residual spatial effects but was not layer-associated ("spatial-associated"), and 512 genes whose expression was layer-associated and also had residual spatial effects ("both"). We found that the enriched GO terms in the three sets of genes had apparent differences (Fig. 6B). In the layer-associated genes, many enriched GO terms were related to synaptic signaling and transmembrane transport; in the spatial-associated genes, most enriched GO terms were related to immune response and cell activation; in genes that were both layer- and spatial-associated, there were also enriched terms relevant to neuron development and differentiation and system process, such as axon development and glial cell differentiation (Fig. 6C and Additional file 1: Fig. S15).

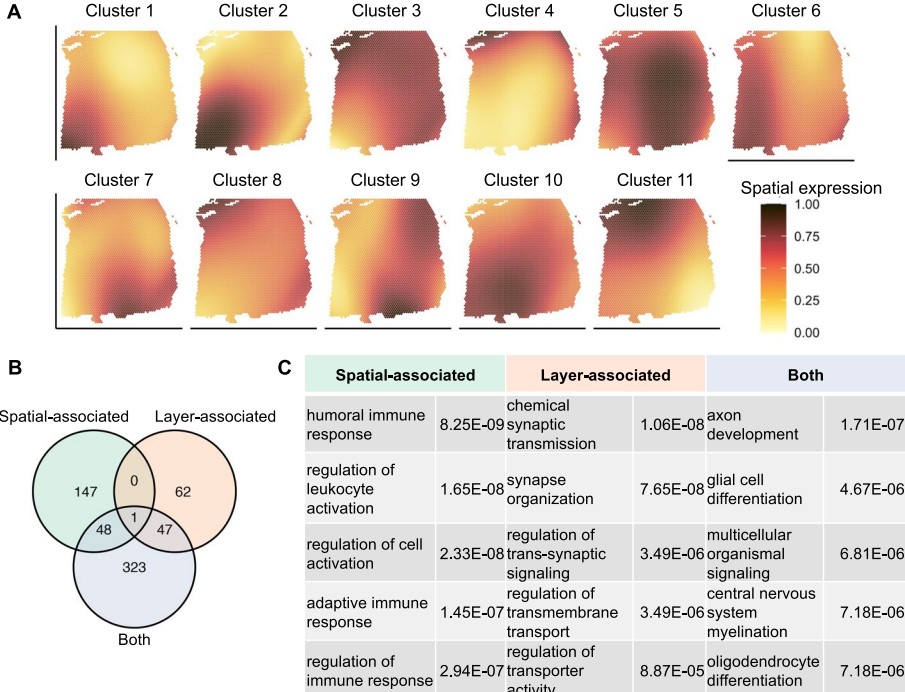

**Fig. 6** spVC's estimation results on the prefrontal cortex spatial transcriptomics data. **A** 11 gene clusters identified from genes with significant residual spatial effects identified by spVC. For every cluster, the residual spatial effects ($\hat{\gamma}_0(\cdot)$) were scaled by the min-max normalization, and the the average was taken across all genes in the cluster to obtain the average spatial expression. **B** Venn diagram of enriched biological process GO terms in the three sets of genes identified by spVC. **C** Selected enriched GO terms in the three sets of genes and their corresponding adjusted *P* values

## Application of spVC to mouse cerebellum data

For the second real data application, we applied spVC to spatial transcriptomics data of mouse cerebellum sequenced by the Slide-seqV2 technology [18, 22]. After filtering genes that were detected in fewer than 100 spots, this dataset included 8213 genes and 11,626 spatial spots. As the gene expression measurements in a spatial spot may reflect a mixture of multiple cells belonging to different cell types, spVC used the estimated proportions of six major cell types (granule, oligodendrocytes, astrocytes, molecular layer interneurons, Bergmann, and Purkinje) by the RCTD method [18] as continuous covariates in this application (see the "Methods" section and Fig. 7A).

On this dataset, we also performed a permutation analysis by randomly permuting the spatial coordinates of the spots, in order to evaluate the type I errors of different methods in detecting residual spatial effects given covariate information. Using 0.05 as a threshold on the adjusted *P* values, we found that spVC had a type I error of 0.001 on this dataset. For comparison, the type I error rates of SpatialDE, SPARK, SPARK-X, and MERINGUE were 0.286, 0.001, 0.345, and 0.004, respectively (Fig. 7B). Like in the previous analysis, we repeated the permutation procedure 100 times and obtained 100 *P* values for each gene. The quantile-quantile plots of the *P* values for two example genes are shown in Additional file 1: Fig. S7C-D. The average RMSE was 0.014, confirming the accuracy of spVC again.

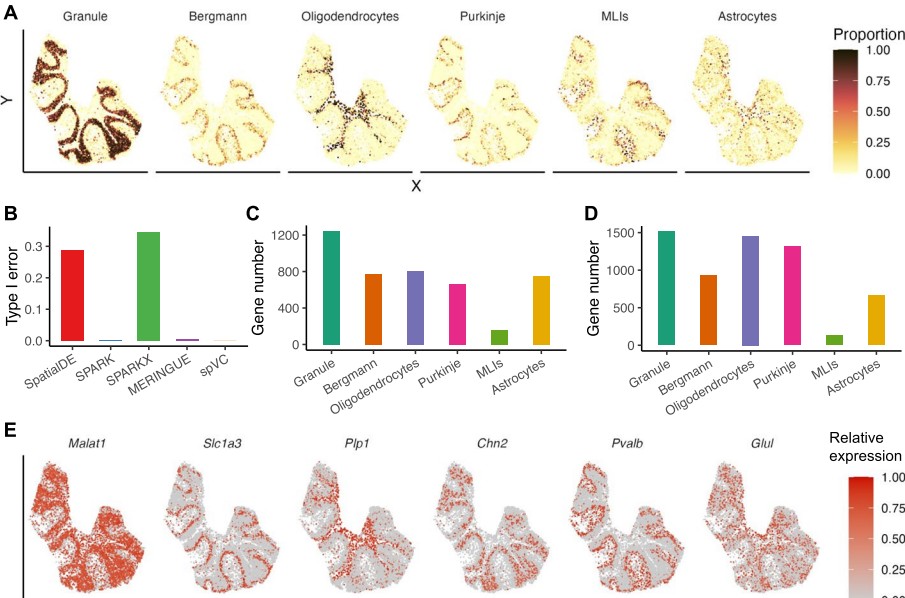

**Fig. 7** spVC's estimation results on the cerebellum spatial transcriptomics data. **A** Cell type proportions of granule cells, Bergmann cells, oligodendrocytes, Purkinje cells, molecular layer interneurons (MLIs), and astrocytes. **B** Type I errors of SpatialDE, SPARK, SPARK-X, MERINGUE, and spVC in the permutation analysis. **C** Number of significant genes whose expression was positively associated with the proportion of each of the six cell types. **D** Number of significant genes whose expression was negatively associated with the proportion of each of the six cell types. **E** Relative expression levels of six example genes positively associated with granule cells, Bergmann cells, oligodendrocytes, Purkinje cells, molecular layer interneurons (MLIs), and astrocytes, respectively. The read counts were normalized by library size, log-transformed, and then scaled by the min-max normalization to obtain the relative expression levels

Next, we applied spVC to the real data to identify cell-type-associated genes. Using 0.05 as a threshold on adjusted *P* values of constant covariate coefficients, we discovered 4759 genes whose expression levels were cell-type-associated and had a significant constant coefficient on at least one of the six cell types (Fig. 7C–D). For example, *Malat1*, *Slc1a3*, *Plp1*, *Chn2*, *Pvalb*, and *Glul* had significantly higher expression in granule, Bergmann, oligodendrocyte, Purkinje, molecular layer interneurons (MLIs), and astrocyte cells, respectively (Fig. 7E). Other top genes with the greatest statistical significance also presented cell-type-specific or cell-type-associated expression patterns (Additional file 1: Fig. S16).

We compared the cell-type-associated genes identified by spVC with the differentially expressed marker genes identified from an snRNA-seq dataset of the mouse cerebellum, which has a median transcript capture of 2862 unique molecular identifiers (UMIs) per cell [23]. In contrast, the Slide-seq data used by spVC only has a median transcript capture of 329 UMIs per spot. Among the 3639 differentially expressed markers detected in both datasets, the numbers of genes upregulated in granule, Bergmann, oligodendrocyte, Purkinje, MLI, and astrocyte cells were 680, 837, 893, 2227, 742, and 783, respectively. In these cell types, the proportion of genes overlapping with those identified by spVC were 54.1% (368), 45.9% (384), 49.2% (439), 24.8% (553), 13.3% (99), and 51.7% (405), respectively. These overlapping genes demonstrated differential expression patterns associated with the corresponding cell type proportions (Additional file 1: Fig. S17). We then evaluated the 2605 genes that were only identified from the snRNA-seq data and the 1676 genes that were only reported by spVC. The genes from snRNA-seq data did not present obviously upregulated expression levels in the claimed cell types (Additional file 1: Fig. S18A), while the genes found by spVC presented expression patterns that were similar to those of the overlapping genes (Additional file 1: Fig. S18B), demonstrating the sensitivity of spVC in identifying cell-type-associated genes from sparse spatial transcriptomics data.

In addition to the cell-type-associated genes, we also identified 304 genes that had significant residual spatial patterns. We compared the deviance of spVC models with and without the spatial effects. For these 304 significant genes, considering the spatial effects led to an apparent decrease of model deviance, compared with the 7909 genes without significant residual spatial patterns (Additional file 1: Fig. S19). For example, spVC found that *Fxyd6* had a significant residual spatial pattern with an adjusted *P* value of $3.89 \times 10^{-4}$ (Fig. 8A–B). Its expression had spatial effects that could not be explained by the varying cell type proportions, and the localization of *Fxyd6* in granule cells and cerebellum has been previously reported [24]. Another two examples were the *Sparc* gene (adjusted $P = 1.87 \times 10^{-13}$), which has potential roles in mediating the inflammation and repair processes of the central nervous system [25], and the *Ttr* gene (adjusted $P = 1.48 \times 10^{-13}$), which is involved in the distribution of thyroid hormones and has protective roles during neurological strokes [26]. The estimated spatial effects ($\hat{\gamma}_0(\cdot)$) by spVC clearly reflected how their expression levels were associated with the spatial coordinates (Fig. 8A–B).

We then tested which genes were subject to spatially varying effects of the cell type proportions, and found 87 genes with significant spatially varying effects of at least one cell type (Fig. 8C). We compared the deviance of spVC models with and without the

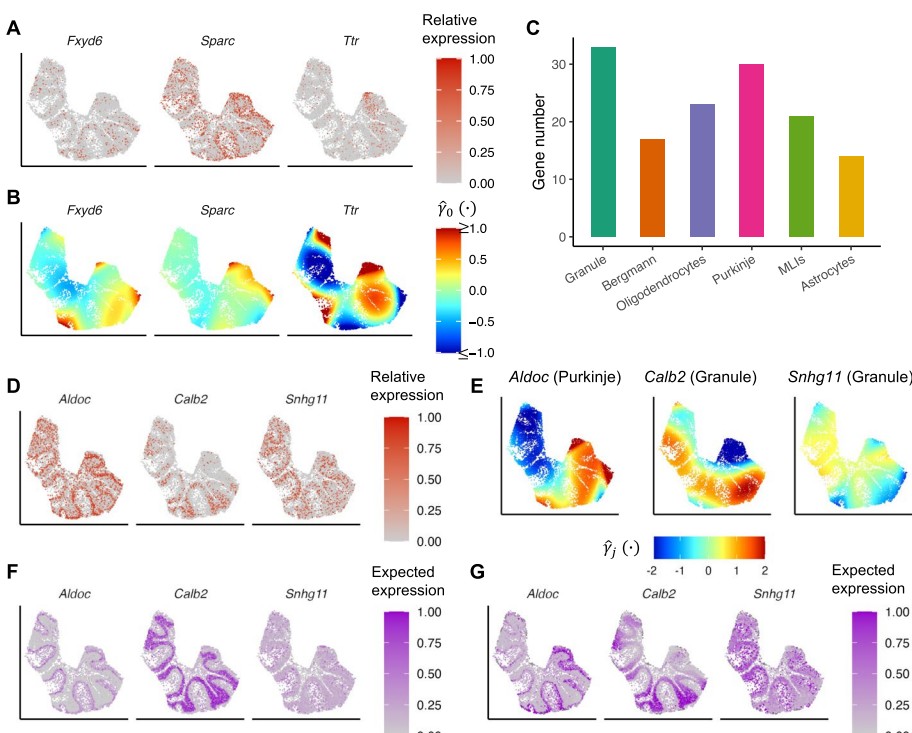

**Fig. 8** Spatially variable genes identified by spVC in the cerebellum spatial transcriptomics data. **A** Relative expression levels of *Fxyd6*, *Sparc*, and *Ttr*. The read counts were normalized by library size, log-transformed, and then scaled by the min-max normalization to obtain the relative expression levels. **B** Estimated residual spatial effects ($\hat{\gamma}_0(\cdot)$) of *Fxyd6*, *Sparc*, and *Ttr*. **C** Number of genes with significant spatially varying effects of each of the six cell types. **D** Relative expression levels of *Aldoc*, *Calb2*, and *Snhg11*. **E** Estimated spatially varying effects of cell type proportions for *Aldoc* (Purkinje cells), *Calb2* (granule cells), and *Snhg11* (granule cells). **F** Expected gene expression of *Aldoc*, *Calb2*, and *Snhg11* estimated by the reduced model which did not consider the spatially varying coefficients of cell type proportions. The shown values were scaled by the min-max normalization. **G** Expected gene expression of *Aldoc*, *Calb2*, and *Snhg11* estimated by the full spVC model. The shown values were scaled by the min-max normalization

spatially varying effects of cell types (see the "Methods" section). For the 87 significant genes, considering the spatially varying effects led to an apparent decrease of model deviance, compared with the 217 genes that had significant residual spatial patterns but without spatially varying effects of cell types (Additional file 1: Fig. S20).

We use three genes as examples to demonstrate the identified spatially varying effects of the cell type proportions (Fig. 8D–E). The first example gene is *Aldoc*, which encodes a glycolytic enzyme and is known to be expressed in subpopulations of Purkinje cells [27]. The observed expression of *Aldoc* presented obvious variation in different lobules. Based on spVC's inference, we found that its expression was subject to spatially varying effects of three cell types: granule cells (adjusted $P = 2.38 \times 10^{-6}$), Purkinje cells (adjusted $P = 2.01 \times 10^{-15}$), and MLIs (adjusted $P = 1.20 \times 10^{-4}$). In particular, the estimated spatial varying effects of Purkinje cell proportions precisely reflected the variation of *Aldoc* expression among different lobules, confirming previous observation that *Aldoc* is expressed heterogeneously in Purkinje cells [27]. The second example gene is *Calb2*, which encodes an intracellular calcium-binding protein. It is expressed in granule cells and modulates intrinsic neuronal excitability [28]. Based on spVC's inference, we found that the expression of *Calb2* was dependent on

spatially varying effects of two cell types: granule cells (adjusted $P = 1.63 \times 10^{-15}$) and oligodendrocytes (adjusted $P = 2.33 \times 10^{-2}$). Its expression variation in different lobules could be largely explained by the spatially varying coefficients of granule cell proportions (Fig. 8E). The third example is *Snhg11*, which encodes a long noncoding RNA shown to be important for synaptic function [29]. Its expression had spatially varying effects of three cell types: granule cells (adjusted $P = 1.90 \times 10^{-7}$), Purkinje cells (adjusted $P = 6.39 \times 10^{-5}$) and MLIs (adjusted $P = 3.11 \times 10^{-4}$). Similar as *Calb2*, the observed expression variation of *Snhg11* in different lobules were explained by the spatially varying coefficients of granule cell proportions (Fig. 8E). In addition, we also compared the fitted expected expression of the three genes by the full spVC model and the reduced model without considering the spatially varying coefficients of cell type proportions. The full spVC model provided a much better fitting of the spatial expression than the reduced model (Fig. 8F–G), highlighting the advantages of incorporating covariates' spatially varying effects in explaining spatial transcriptomics data.

We also performed a GO enrichment analysis to investigate the biological processes represented by the identified genes. We considered two gene sets, the 4463 genes which had a significant constant coefficient on at least one of the six cell types but did not have significant residual spatial patterns ("cell-type-associated"), and the 304 genes which had significant residual spatial patterns ("spatial-associated"). Using 0.05 as a threshold on adjusted $P$ values, we found 745 and 268 enriched GO terms in the cell-type-associated and spatial-associated genes, respectively, and 152 of these GO terms were shared. Among the 593 GO terms that were only enriched in the cell-type-associated genes, we found biological processes that were related to cellular component organization, cell adhesion, and nervous system development, such as gliogenesis, oligodendrocyte differentiation, and astrocyte differentiation (Additional file 1: Table S1). Among the 116 GO terms that were only enriched in the spatial-associated genes, many top GO terms were relevant to transmembrane transport and regulation of transport (Additional file 1: Table S2). In addition, multiple enriched processes were related to muscle contraction and regulation of muscle contraction, suggesting that the spatial-associated genes play key roles in cerebellum's function of motor movement and balance control [30]. Lastly, among the 152 GO terms that were enriched in both gene sets, top terms were related to synaptic signaling and transmembrane transport (Additional file 1: Table S3).

Lastly, we conducted a comparison between the spatially variable genes identified by spVC and those identified by alternative methods. After adjusting for the cell type proportions, spVC identified 304 genes with significant residual spatial effects, while SpatialDE, SPARK, SPARK-X, and MERINGUE identified 2183, 820, 3818, and 278 genes, respectively. We found that 96.4% of spVC's spatial genes were identified by at least one other method. However, the proportion of overlap for SpatialDE, SPARK, SPARK-X, and MERINGUE reduced to 44.2%, 84.8%, 34.2%, and 80.2%, respectively. We also found that a large proportion of genes reported to have significant residual spatial patterns by other methods were only found to be cell-type-associated by spVC (SpatialDE: 44.5%; SPARK: 58.5%; SPARK-X: 63.0%; MERINGUE: 46.8%; Additional file 1: Table S4). Among these genes, we visualized the top ones by $P$ values, and didn't observe obvious spatial patterns not explained by the cell type proportions (Additional file 1: Fig. S21).

## Application of spVC to mouse testis data

To further investigate spVC's potential in uncovering spatially varying effects of covariates on gene expression, we applied spVC to a mouse testis dataset [31] obtained using the Slide-seq technology. After filtering, this dataset comprised 10,527 genes and 18,097 spots (Methods). In order to analyze the spatio-temporal gene expression regulation, we used the pseudotime orders inferred by Chen et al. [31] as a continuous covariate in this application. These pseudotime orders confirmed the recognized developmental trajectory of germ cells, which starts from the basement membrane and progresses towards the lumen of the seminiferous tubules (Fig. 9A).

Using spVC and 0.05 as a threshold on adjusted *P* values, we first identified 8998 genes with a significant constant coefficient of the pseudotime orders. Among these, 1370 genes exhibited increased expression in the trajectory of germ cell development, while the remaining 7628 genes showed decreased expression. For example, the *Smcp* gene, which encodes a sperm mitochondria-associated cysteine-rich protein, had a positive coefficient of 3.79 (adjusted $P \approx 0$), indicating higher expression in the later stage of the germ cell development trajectory as confirmed by single-molecule fluorescence *in situ* hybridization (smFISH) image [31] (Fig. 9B). Conversely, the *Lyar* gene, which encodes a cell growth-regulating nucleolar protein, had a negative coefficient of −2.91 (adjusted $P \approx 0$), and was shown to have higher expression in the earlier stage of the development trajectory by smFISH image (Fig. 9B). Subsequently, we identified 546 genes that presented significant spatially varying coefficients of the pseudotime orders, suggesting

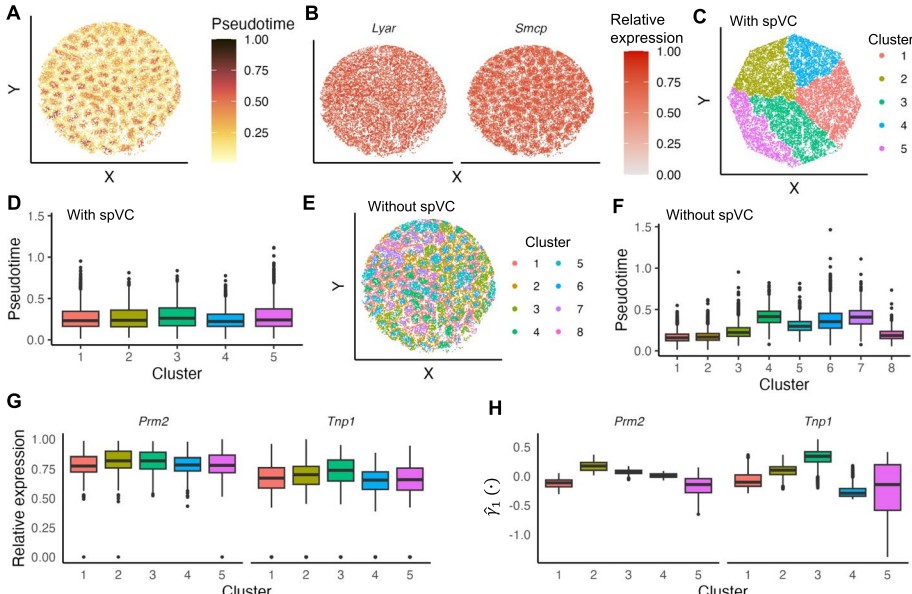

**Fig. 9** Application of spVC to the mouse testis dataset. **A** Pseudotime values of the observed spatial spots. **B** Relative expression levels of *Lyar* and *Smcp*. The read counts were normalized by library size, log-transformed, and then scaled by the min-max normalization to obtain the relative expression levels. **C** Five spot clusters identified based on the spatially varying effects ($\gamma_1(\cdot)$) of pseudotime inferred by spVC. **D** Distribution of pseudotime values within the five clusters shown in **C**. **E**: Eight spot clusters identified based on the observed gene expression levels. **F** Distribution of pseudotime values within the eight clusters shown in **E**. **G** Relative expression levels of *Prm2* and *Tnp1* within the five clusters shown in **C**. **H** Estimated spatially varying effects of *Prm2* and *Tnp1* within the five clusters shown in **C**

a dynamic interplay between spatial and temporal dimensions. To further understand these identified spatial effects, we clustered the spots based on their corresponding spatially varying coefficients ($\gamma_1(\cdot)$) of the 546 genes, and obtained five spot clusters (Fig. 9C; Methods). The median pseudotime of these clusters was similar, indicating that the clustering analysis based on spVC's estimated spatial coefficients was not confounded by the actual pseudotime values of the spots (Fig. 9D). In contrast, direct clustering analysis using observed gene expression yielded eight spot clusters with apparent differences in their pseudotime values, suggesting that the clustering process was primarily driven by expression changes along the germ cell development trajectory (Fig. 9E–F).

We further scrutinized the five spot clusters unveiled by spVC and observed significant differences in genes previously known to be associated with various stages of the epithelium cycle of seminiferous tubules [31, 32], including *Prm1* ($P \approx 0$), *Tnp1* ($4.68 \times 10^{-192}$), *Prm2* ($4.57 \times 10^{-122}$), *H3f3b* ($5.11 \times 10^{-64}$), *Habp4* ($1.84 \times 10^{-18}$), *Trim24* ($4.68 \times 10^{-4}$), *Smarca2* ($6.54 \times 10^{-4}$), and *Ezh2* ($9.30 \times 10^{-4}$). The *P* values were obtained by conducting the Kruskal-Wallis test on relative gene expression levels followed by the BH correction. For example, *Prm1* has been associated with step 15 spermatids onward, while *Tnp1* has been linked to spermatids of steps 9–12 [32]. In addition to presenting significant difference in observed gene expression, these two genes exhibited more distinct cluster-dependent patterns in the spatial coefficients estimated by spVC (Fig. 9G–H). The germ cell development cycle and the seminiferous epithelium cycle are two interrelated yet distinct processes occurring within the testes, working together to ensure a continuous production of spermatozoa. Our analysis demonstrates the potential of spVC to unveil spot clusters representing different stages of the seminiferous epithelium cycle by estimating the spatially varying effects of spot pseudotimes and characterizing the spatio-temporal dynamics of gene expression during spermatogenesis. This potential is obscured when analyzing observed gene expression alone.

## Discussion

In this article, we propose a statistical method named spVC for analyzing and interpreting spatial variation of gene expression in spatial transcriptomics data. The spVC method provides a convenient tool to identify potential factors that contribute to gene expression variability, including spatial locations and other cell/spot-level covariates such as cell types or cell states. The flexible statistical model also enables the quantification of constant or spatially varying effects of covariates across the studied spatial domain. Our simulation study demonstrates that spVC can effectively distinguish between spatial variation introduced by covariate effects and residual spatial effects which cannot be explained by available covariates.

The spVC method not only performs the detection of SVGs but also integrates information from cell/spot-level covariates to untangle interactions between spatial locations and other factors that govern gene expression. For instance, when applied to the human cortex data, spVC identified diverse types of SVGs. These include layer-associated genes, where observed spatial variation arose solely from the distribution pattern of different cortex layers; spatial-associated genes, whose expression did not differ between cortex layers but was directly dependent on spatial coordinates; and genes showing spatial variation resulting from a combination of layer-specific expression and unexplained spatial effects. In the

context of the mouse cerebellum data, alongside the discovery of cell-type-associated genes and spatial-associated genes, spVC revealed a group of genes subject to spatially varying effects of cell type proportions. While our real-data applications utilized cellular layers or cell type proportions as spot-level covariates, it is noteworthy that the spVC model itself does not constrain the types of covariates that can be incorporated into the analysis. By inputting various factors into the model, we anticipate that spVC will address previously unapproachable questions related to transcriptional regulation in the spatial context. Meanwhile, we would like to point out that the current spVC model treats the covariate information as given and fixed, so potential estimation or measurement errors contained in the covariates are not modeled in spVC. Therefore, if the covariates require computational inference, we recommend utilizing tools that have demonstrated good performance in previous benchmark studies.

We would like to discuss several methods that address pertinent questions while emphasizing different computational challenges from those considered by spVC. First, while the objective of spVC is to examine the spatial variation of gene expression from spatial transcriptomics data (and available covariates), the SPADE method [33] aims to identify marker genes through a combined analysis of transcriptomics data and corresponding histology images. Our previous benchmark study indicates that the inclusion of histology information does not always enhance clustering analysis of spatial transcriptomics data [7]. However, it remains valuable to explore how histology data contributes to the identification and comprehension of SVGs. Second, spVC diverges from SVG detection methods relying on differential expression tests [34]. These methods primarily focus on identifying genes that are enriched within a particular spatial subdomain in contrast to other subdomains. Third, when cell type labels or proportions are integrated as covariates in spVC, it can investigate interactions between cell types and spatial locations in the context of gene expression regulation. In camparison, the CTSV method [35] examines whether genes are spatially variable within individual cell types, assuming spatial patterns to be linear, squared exponential, or periodic; the C-SIDE method [36] also uses cell type proportions to infer differential expression in spatial data. However, unlike spVC, these two methods do not attempt to decompose spatial variation into effects explainable by covariates and unexplained residual spatial effects. Fourth, we have highlighted that an important feature of spVC is its ability to estimate the expected covariate effects and spatial effects on gene expression in a designated spatial domain. A recent method named GASTON [37] is able to characterize spatial expression variation using piece-wise linear expression functions, while it does not aim to perform inference on covariate-associated and residual spatial patterns.

In summary, spVC expands the analysis of spatial transcriptomics data by simultaneously achieving the detection of SVGs and the decomposition of spatial expression variation into covariate effects (constant or spatially varying) and residual spatial effects. We anticipate that spVC will be a useful tool in spatial transcriptomics and provide new perspectives on the spatial variation and regulation of gene expression.

## Conclusions

In this article, we introduce spVC, a novel statistical method to detect and interpret SVGs based on a generalized Poisson model. spVC provides a convenient tool to identify potential factors that contribute to gene expression variability, including spatial

locations and other cell/spot-level covariates such as cell types or tissue layers. It offers estimation and statistical inference tools for both constant and spatially varying coefficients, allowing for the selection of different types of SVGs. Our simulation and real data applications have demonstrated spVC's accuracy in the above tasks. In summary, spVC is a versatile tool for the identification, interpretation, and comprehension of gene expression variation in spatial transcriptomics data.

## Methods

### The spVC model

In this section, we introduce the spVC model, which is a generalized regression model that allows spatially varying coefficients of explanatory variables (Fig. 2A). For simplicity of notations, we focus the discussion on spatial transcriptomics data with two-dimensional spatial locations, but the statistical model is also applicable to three-dimensional data. Let $s_i = (s_{i1}, s_{i2})^\top$ be the location of the $i$th cell (or spot), $i = 1, \ldots, n$, which belongs to a bounded domain $\Omega \subseteq \mathbb{R}^2$ of an arbitrary shape. At location $s_i$, $Y_i$ is the count of RNA-seq reads, $x_i = (x_{i1}, \ldots, x_{ip})^\top$ is a vector of cell/spot-level covariates which may have different effects on the gene expression levels. These covariates will be selected by the users depending on the biological questions. They can be discrete or continuous. In the context of spatial transcriptomics data, these variables can be cell type indicators, cell type abundances in the cells' neighborhood region, abundance levels of known regulatory factors, etc. We assume the conditional density of $Y_i$ given $(x_i, s_i)$ follows a Poisson distribution $f_{Y_i|x_i,s_i}(y|x,s) = \text{Poisson}(\mu(x,s))$, where $\mu(x,s) = \text{E}(Y|x,s)$ is the expected gene expression given $(x, s)$. In this work, $\mu(x, s)$ is modeled via a link function $g = \log(\cdot)$ as follows:

$$g\{\mu(x_i, s_i)\} = \log \ell_i + \beta_0 + \gamma_0(s) + \sum_{j=1}^{p} x_{ij}\beta_j + \sum_{j=1}^{p} x_{ij}\gamma_j(s), \qquad (1)$$

where $\boldsymbol{\beta} = (\beta_0, \beta_1, \ldots, \beta_p)^\top$ is a vector of unknown constants, which describes the intercept and spatially constant effects of the covariates. The coefficient functions $\boldsymbol{\Gamma}(s) = \{\gamma_0(s), \gamma_1(s), \ldots, \gamma_p(s)\}$ describes the spatially varying effects. More specifically, $\gamma_j(s)$ describes the spatially varying effect of covariate $j$ ($j = 1, \ldots, p$), and $\gamma_0(s)$ describe the spatial effects that cannot be explained by any known covariates. We refer to $\gamma_0(s)$ as the residual spatial effect. For the model's identifiability, we require that $\int_\Omega \gamma_j(s)ds = 0$, $j = 0, 1, \ldots, p$. We use $\ell_i$ to denote the library size factor of the $i$th cell/spot, and it is defined as the cell/spot's library size divided by the median library size across all cells/spots. The above model is constructed for each gene independently. It is worth noting that the Poisson distribution is widely used to model spatial transcriptomics data [16, 38]. We introduce how over-dispersion is accounted for in our method in model estimation below.

The proposed spVC model is able to account for cell-level or spot-level covariates with potentially various effects on gene expression. We discuss several typical scenarios below. (1) If $\beta_j = 0$ and $\gamma_j(s) \equiv 0$ for $j = 1, \ldots, p$, then none of the covariates has effect on gene expression, and any spatial pattern, if exists, is reflected in the residual spatial effect $\gamma_0(s)$. (2) If for any $j \in \{1, \ldots, p\}$, $\beta_j \neq 0$ and $\gamma_j(s) \equiv 0$, then the $j$th covariate has

a constant (i.e., spatially invariant) effect on gene expression. (3) If for any $j \in \{1, \ldots, p\}$, $\gamma_j(\boldsymbol{s}) \not\equiv 0$, then the $j$th covariate has a spatially varying effect on gene expression. It is worth noting that, when a covariate only takes binary values, and the spatial spots where the covariate takes values of ones are clustered (in contrast to dispersed), then it is not feasible to consider the spatially varying effect of the covariate. In this case, the spatially varying effect of the covariate, if exists, will not be identifiable from the residual spatial effect. For instance, in the application on the human cortex data, the spots were annotated by seven layers (six neocortical layers and the white matter). The covariates were binary labels for the six neocortical layers, which had clear spatial boundaries (Fig. 5A). Thus, we did not consider the spatially varying effects of covariates in this application.

### *Model estimation*

Since spatial transcriptomics data is often over-dispersed, we utilize the quasi-likelihood [39], a standard approach in generalized linear models allowing for overdispersion, to estimate the model (1). To define a quasi-likelihood function, we only need to specify a relationship between the mean and variance of the responses. Note that the observed count of cell/spot $i$, $Y_i$, has a mean of $\mu_i = \mu(\boldsymbol{x}_i, \boldsymbol{s}_i)$. We specify $\mathrm{Var}(Y|\boldsymbol{x}, \boldsymbol{s}) = \phi V\{\mu(\boldsymbol{x}, \boldsymbol{s})\}$, which links the variability in the response variable $Y$ to its mean, given the predictors $\boldsymbol{x}$ and spatial location $\boldsymbol{s}$. In this work, $\phi$ is the unknown parameter for overdispersion and the variance function follows $V\{\mu(\boldsymbol{x}, \boldsymbol{s})\} = \mu(\boldsymbol{x}, \boldsymbol{s})$ for the Poisson distribution. Then, we define a quasi-likelihood function $h(\mu, y)$ which satisfies that $\nabla_\mu h(\mu, y) = (y - \mu)/\{\phi V(\mu)\}$, where $\nabla_\mu$ denotes the partial derivative. The quasi-likelihood estimators of $\boldsymbol{\beta}$ and $\boldsymbol{\Gamma}(\boldsymbol{s})$ maximize the objective function $L = \sum_{i=1}^n h\big[g^{-1}\{\mu(\boldsymbol{x}_i, \boldsymbol{s}_i)\}, Y_i\big]$, where $\mu(\boldsymbol{x}_i, \boldsymbol{s}_i)$ is specified in model (1).

To approximate the spatially varying coefficient functions $\gamma_j(\boldsymbol{s})$, $j = 0, \ldots, p$, we consider the bivariate penalized spline over triangulation (BPST) method (Additional file 2: Supplementary methods) [40]. BPST first uses a triangle mesh to approximate the irregular spatial domain. For domain $\Omega$ from which the spatial transcriptomics data is generated, we consider the triangulation partition $\triangle = \{T_1, \ldots, T_K\}$ of $K$ triangles. For example, the triangulation partitions for the human cortex data and mouse cerebellum data discussed in real data applications are presented in Additional file 1: Figs. S22 and S23. We use $B_m(\boldsymbol{s})$ ($m \in \mathcal{M}$) to denote the centered spline basis functions with $\int_\Omega B_m(\boldsymbol{s})d\boldsymbol{s} = 0$, and $\mathcal{M}$ is the index set of basis functions. For any $\boldsymbol{s} \in \Omega$, $\mathbf{B}(\boldsymbol{s}) = (B_1(\boldsymbol{s}), \ldots, B_M(\boldsymbol{s}))^T$ denotes the vector of spline basis functions evaluated at $\boldsymbol{s}$. Then, the bivariate function $\gamma_j(\boldsymbol{s})$ can be approximated by a linear combination of the spline basis functions $\gamma_j(\boldsymbol{s}) \approx \mathbf{B}(\boldsymbol{s})^\top \boldsymbol{\theta}_j$, subject to $\mathbf{H}\boldsymbol{\theta}_j = \mathbf{0}$ ($\forall j \in \{0, 1, \ldots, p\}$). The condition on the spline coefficients ($\mathbf{H}\boldsymbol{\theta}_j = \mathbf{0}$) guarantees that $\mathbf{B}(\boldsymbol{s})^\top \boldsymbol{\theta}_j$ is a smooth function with continuous first-order derivative, where the matrix $\mathbf{H}$ embeds the smoothness assumptions across all the shared edges of triangles. The dimensions of $\mathbf{H}$ are $J \times |\mathcal{M}|$, where $J$ depends on the triangulation structure (i.e., the number of shared edges) and the smoothness assumption. See [41] for the detailed construction of basis functions and [42] for the construction of the $\mathbf{H}$ matrix.

By incorporating the BPST method, we obtain the quasi-likelihood estimators, $(\widehat{\beta}_j, \widehat{\boldsymbol{\theta}}_j)$ ($j = 0, \ldots, p$), by maximizing

$$L_{\text{BPST}} = \sum_{i=1}^{n} h\left[ g^{-1}\left\{ \log \ell_i + \sum_{j=0}^{p} x_{ij}\beta_j + \sum_{j=0}^{p} x_{ij}\mathbf{B}(\boldsymbol{s}_i)^{\top}\boldsymbol{\theta}_j \right\}, Y_i \right] - \frac{1}{2}\sum_{j=0}^{p} \lambda_j \mathcal{E}\{\mathbf{B}^{\top}\boldsymbol{\theta}_j\},$$

(2)

where $x_{i0} \triangleq 1$ and $\mathbf{H}\boldsymbol{\theta}_j = \mathbf{0}$. $\mathcal{E}(\cdot)$ is a roughness penalty term with $\mathcal{E}(f) = \int_{\Omega}\{(\nabla_{s_1}^2 f)^2 + 2(\nabla_{s_1}\nabla_{s_2}f)^2 + (\nabla_{s_2}^2 f)^2\}ds_1 ds_2$. Therefore, the problem is equivalent to maximizing

$$L^P(\boldsymbol{\beta},\boldsymbol{\theta}) = \sum_{i=1}^{n} h\left[ g^{-1}\left\{ \log \ell_i + \sum_{j=0}^{p} x_{ij}\beta_j + \sum_{j=0}^{p} x_{ij}\mathbf{B}(\boldsymbol{s}_i)^{\top}\boldsymbol{\theta}_j \right\}, Y_i \right] - \frac{1}{2}\sum_{j=0}^{p} \lambda_j \boldsymbol{\theta}_j^{\top}\mathbf{P}\boldsymbol{\theta}_j,$$

(3)

subject to $\mathbf{H}\boldsymbol{\theta}_j = \mathbf{0}$, and $\mathbf{P}$ is a block diagonal penalty matrix satisfying $\boldsymbol{\theta}_j^{\top}\mathbf{P}\boldsymbol{\theta}_j = \mathcal{E}\{\mathbf{B}^{\top}\boldsymbol{\theta}_j\}$ ($\forall j \in \{0, 1, \ldots, p\}$). Details about the construction of matrix $\mathbf{P}$ are available in the Supplementary Information (Section B.2.2) in [42]. We use iterative reweighted least squares to solve the objective function in formula (3) and use the generalized cross-validation (GCV) to select the regularization parameters ($\lambda_j$) [43]. Once the quasi-likelihood estimators are obtained, the bivariate varying coefficient functions can be estimated by $\widehat{\gamma}_j(\boldsymbol{s}) = \mathbf{B}(\boldsymbol{s})^{\top}\widehat{\boldsymbol{\theta}}_j$ ($j = 0, \ldots, p$). In addition, the overdispersion parameter is estimated by $\widehat{\phi} = \frac{1}{n}\sum_{i=1}^{n}\{Y_i - g^{-1}(\widehat{\eta}_i)\}^2/V(g^{-1}(\widehat{\eta}_i))$, where $\widehat{\eta}_i = \log \ell_i + \sum_{j=0}^{p} x_{ij}\widehat{\beta}_j + \sum_{j=0}^{p} x_{ij}\mathbf{B}(\boldsymbol{s}_i)^{\top}\widehat{\boldsymbol{\theta}}_j$. We have illustrated the main steps of the model estimation in Fig. 2B.

### Hypothesis testing

As a statistical framework, spVC accommodates two testing procedures based on users' preferences. Below, we introduce a two-step testing procedure (Fig. 2C–D), which is a common practice in regression modeling. In addition to this approach, the spVC software also facilitates a comprehensive testing procedure, which directly tests all components using the full model.

For each gene, in the first step, we test the significance of the spatially invariant effects of the covariates $\boldsymbol{\beta} = (\beta_0, \beta_1, \ldots, \beta_p)^{\top}$ and the residual spatial effect $\gamma_0(\cdot)$ through the following reduced model:

$$g\{\mu(\boldsymbol{x}_i, \boldsymbol{s}_i)\} = \log \ell_i + \beta_0 + \gamma_0(\boldsymbol{s}_i) + \sum_{j=1}^{p} x_{ij}\beta_j.$$

(4)

In order to detect the spatially invariant effects of the explanatory variables, we consider the following hypothesis test for the $j$th covariate ($j \in \{1, \ldots, p\}$):

$$H_{0j}: \quad \beta_j = 0, \quad H_{Aj}: \quad \beta_j \neq 0.$$

(5)

In order to detect the residual spatial effect, we consider the hypothesis test:

$$H_0^s: \quad \gamma_0(\boldsymbol{s}) = 0, \text{ for all } \boldsymbol{s} \in \Omega, \quad H_A^s: \quad \gamma_0(\boldsymbol{s}) \neq 0, \text{ for some } \boldsymbol{s} \in \Omega.$$

(6)

In the second step, we test the significance of the spatially varying effects using model (1) (the full model). We only consider the spatially varying effect of a covariate if the

constant effect of this variable is significant and the gene demonstrates significant residual spatial patterns in the reduced model. In other words, a gene will be tested in the second step only if $H_0^s$ and at least one $H_{0j}$ ($j = 1, \ldots, p$) are rejected. In order to detect the spatially varying effects, we consider the following hypothesis test for the $j$th variable:

$$H_{0j}^s: \quad \gamma_j(\boldsymbol{s}) = 0, \text{ for all } \boldsymbol{s} \in \Omega, \quad H_{Aj}^s: \quad \gamma_j(\boldsymbol{s}) \neq 0, \text{ for some } \boldsymbol{s} \in \Omega. \tag{7}$$

To conduct the hypothesis tests in (5), we use a likelihood ratio test through the ANOVA table generated from the *mgcv* package [44]. The corresponding test statistic is the deviance difference between the full and reduced models. To conduct the hypothesis tests in (6) and (7), we use a Wald-type test, whose general framework was first proposed in [45]. Suppose we use $\boldsymbol{1}$ to denote a vector of ones, $\boldsymbol{e}_j$ to denote a vector of length $p + 1$ with the $(j + 1)$th element being one and the other components being zero, $\mathbf{X} = \{\boldsymbol{x}_1, \ldots, \boldsymbol{x}_n\}^\top$, $\mathbf{B} = \{\mathbf{B}(\boldsymbol{s}_1), \ldots, \mathbf{B}(\boldsymbol{s}_n)\}^\top$, and $\boldsymbol{\Lambda} = \text{diag}(\lambda_0, \lambda_1, \ldots, \lambda_p)$. The test statistic is given by

$$T_j^\alpha = \widehat{\boldsymbol{\gamma}}_j^\top \mathbf{V}_j^{\alpha-} \widehat{\boldsymbol{\gamma}}_j,$$

where $\widehat{\boldsymbol{\gamma}}_j = \{\widehat{\gamma}_j(\boldsymbol{s}_i), i = 1, \ldots, n\}$ is a vector of estimated $\widehat{\gamma}_j(\boldsymbol{s})$ at points $\boldsymbol{s}_i$ ($i = 1, \ldots, n$), and $\mathbf{V}_j$ is the estimated covariance matrix of $\widehat{\boldsymbol{\gamma}}_j$. Denote $\otimes$ as the Kronecker product operator. Then, the individual elements in $\mathbf{V}_j = (V_{ii',j})_{1 \leq i, i' \leq n}$ is given by

$$V_{ii',j} = \widehat{\phi}\left(\boldsymbol{0}^\top \quad \mathbf{B}(\boldsymbol{s}_i)^\top \otimes \boldsymbol{e}_j^\top\right)\left(\mathbf{D}^\top \mathbf{W} \mathbf{D} + \begin{pmatrix} \boldsymbol{0} & \boldsymbol{0} \\ \boldsymbol{0} & \boldsymbol{\Lambda} \otimes \mathbf{P} \end{pmatrix}\right)^{-1}\left(\begin{matrix} \boldsymbol{0} \\ \boldsymbol{e}_j \otimes \mathbf{B}(\boldsymbol{s}_{i'}) \end{matrix}\right),$$

where $\widehat{\phi}$ is the estimated over-dispersion parameter, $\mathbf{D}$ is a $n \times (p + 1)(|\mathcal{M}| + 1)$ matrix with $\mathbf{D} = \mathbf{X} \otimes [\mathbf{1} \ \mathbf{B}]$ ($[\mathbf{1} \ \mathbf{B}]$ denotes a matrix by appending $\mathbf{1}$ as a column to $\mathbf{B}$), and $\mathbf{W}$ is a diagonal matrix such that $\mathbf{W}_{ii}^{-1} = \mu_i \times g'(\mu_i)^2$ for Poisson distribution. In addition, $\mathbf{V}_j^{\alpha-}$ is a rank-$\alpha$ pseudo-inverse of the matrix $\mathbf{V}_j$. To choose the value of $\alpha$, we adopt the method of a well-behaved Wald statistic in Section 2.2 of Wood [45]. The parameter $\alpha$ is determined by the matrix $\mathbf{D}\left(\mathbf{D}^\top \mathbf{W} \mathbf{D} + \begin{pmatrix} \boldsymbol{0} & \boldsymbol{0} \\ \boldsymbol{0} & \boldsymbol{\Lambda} \otimes \mathbf{P} \end{pmatrix}\right)^{-1}\mathbf{D}^\top \mathbf{W}$, where the most heavily penalized components are dropped to enhance the testing power. Under the null hypothesis, the asymptotic distributions of the test statistic are derived in [45]. When $\alpha$ is an integer, $T_j^\alpha \sim \chi_\alpha^2$. Otherwise, $T_j^\alpha \sim \chi_{k-2}^2 + \nu_1 \chi_1^2 + \nu_2 \chi_1^2$, where $k = \lfloor \alpha \rfloor$, $\nu = \alpha - k + 1$, $\nu_1 = \{\nu + 1 + (1 - \nu^2)^{1/2}\}/2$ and $\nu_2 = \nu + 1 - \nu_1$.

### Generation of simulated data

In order to evaluate the performance of the spVC model, we designed a simulation study based on the real spatial transcriptomics dataset of mouse cerebellum [18]. Our simulation study considered four groups of genes. For genes in Group 1 and Group 2, there did not exist any spatial effects on gene expression, and these genes were used to evaluate the type I error of the SVG methods. Genes in Group 3 and Group 4 were designed to evaluate the power of the methods when the covariates had constant or spatially varying effects. In Groups 3 and 4, we considered four covariates corresponding to the cell type proportions of four cell types. These values were calculated based on the proportions of Bergmann cells, granule cells, molecular layer interneurons

(MLIs), and oligodendrocytes from the mouse cerebellum data (Additional file 1: Fig. S24). Below we introduce how the simulated data, a read count matrix of $I$ spots ($I = 500, 1000, 2000, 5000,$ or $8000$) and $K = 20,000$ genes, was generated.

We first generated the two-dimensional spatial coordinates $s_i$ of spot $i$ ($i = 1, \ldots, I$) by randomly selecting $I$ data points from the observed locations of the mouse cerebellum data. Next, we generated the expected expression levels of the genes in different spots. We use the parameter $\mu_{ik}$ to represent the expected expression of gene $k$ in spot $i$, and it was directly generated using the following procedures for the four groups of genes. Each group contained 5000 genes.

*Group 1: No covariate effect + No residual spatial effect.* For genes in Group 1, we assumed that the log-transformed expected expression only depended on an intercept:

$$\log\{\mu_{ik}(\boldsymbol{x}_i, \boldsymbol{s}_i)\} = \beta_{0k},$$

where $\beta_{0k}$ was uniformly sampled from $U[-2.5, -2.1]$, based on the observed expression levels of genes detected in at least 5% cells in the real data. Once $\beta_{0k}$'s were generated, $\mu_{ik}$'s could be calculated accordingly.

*Group 2: Constant covariate effect + No residual spatial effect.* For genes in Group 2, we assumed that the log-transformed expected expression depended on the cell type proportions through a linear relationship:

$$\log\{\mu_{ik}(\boldsymbol{x}_i, \boldsymbol{s}_i)\} = \beta_{0k} + \sum_{j=1}^{4} x_{ij}\beta_{jk},$$

where $\beta_{0k}$ was uniformly sampled from $U[-2.5, -2.1]$, and $\beta_{jk}$ ($j = 1, 2, 3, 4$) denoted the spatially constant effect of cell type $j$ on gene $k$. For each gene, we assumed there were two cell type proportions with non-zero coefficients. First, we randomly selected two cell types whose $\beta_{jk}$'s were sampled from $U[1.0, 1.4]$ or $U[-1.4, -1.0]$ with equal probabilities. Then, we set the coefficients of the other two cell types as 0.

*Group 3: Constant covariate effect + Residual spatial effect.* For genes in Group 3, we assumed that the log-transformed expected expression was determined by both cell type proportions and a spatial effect independent of cell type proportions:

$$\log\{\mu_{ik}(\boldsymbol{x}_i, \boldsymbol{s}_i)\} = \beta_{0k} + \gamma_{0k}(\boldsymbol{s}_i) + \sum_{j=1}^{4} x_{ij}\beta_{jk},$$

where $\beta_{0k}$ was uniformly sampled from $U[-2.5, -2.1]$. We used the same way as in Group 2 to randomly generate $\beta_{jk}$. To generate the residual spatial effect, we first independently sampled $\xi_{1k}$ and $\xi_{2k}$ from $U[-3.5, 4]$ and randomly sampled $\delta_{0k}$ from $\{-1, 1\}$ with equal probabilities. Then, we defined a spatial function $\widetilde{\gamma}_{0k}(\boldsymbol{s}) = 2\delta_{0k} \exp\{-0.05(s_1 - \xi_{1k})^2 - 0.05(s_2 - \xi_{2k})^2\}$. Lastly, $\gamma_{0k}(\boldsymbol{s})$ was obtained by centering the function $\widetilde{\gamma}_{0k}(\boldsymbol{s})$ so its average value across all spots was 0.

*Group 4: Spatially varying covariate effect + Residual spatial effect.* For genes in Group 4, we assumed that the log-transformed expected expression was determined by both cell type proportions and a spatial effect independent of cell type proportions; in addition, the effects of cell type proportions ($x_2$ and $x_4$) were spatially varying:

$$\log\{\mu_{ik}(\boldsymbol{x}_i, \boldsymbol{s}_i)\} = \beta_{0k} + \gamma_{0k}(\boldsymbol{s}_i) + \sum_{j=1}^{4} x_{ij}\beta_{jk} + \gamma_{2k}(\boldsymbol{s}_i)x_2 + \gamma_{4k}(\boldsymbol{s}_i)x_4.$$

In Group 4, we used the same approach as in Group 3 to generate $\beta_{j0}, \beta_{j2}, \beta_{j4}$ and $\gamma_{0k}(\boldsymbol{s})$. $\beta_{j1}$ and $\beta_{j3}$ were set to 0. In addition, we assumed that $\widetilde{\gamma}_{2k}(\boldsymbol{s}) = 2\delta_{1k}\cos(s_2 + \xi_{3k})$, $\widetilde{\gamma}_{4k}(\boldsymbol{s}) = 2\delta_{2k}\cos(s_1 + \xi_{4k})$, where $\xi_{3k}$ and $\xi_{4k}$ were uniformly generated from $U[1, 5]$; $\delta_{1k}$ and $\delta_{2k}$ were randomly sampled from $\{-1, 1\}$ with equal probabilities. Lastly, $\gamma_{2k}(\boldsymbol{s})$ and $\gamma_{4k}(\boldsymbol{s})$ were obtained by centering the $\widetilde{\gamma}_{2k}(\boldsymbol{s})$ and $\widetilde{\gamma}_{4k}(\boldsymbol{s})$ functions, respectively.

In order to generate the final count data based on the gene expression parameters obtained above, we used the multinomial distribution to describe the sequencing process. We assumed that, for spot $i$, the read counts of the genes followed a multinomial distribution with a library size of $n_i$ and probabilities $\boldsymbol{p}_i = \{p_{ik}\}_{k=1}^{K}$, where $p_{ik} = \mu_{ik} / \left( \sum_{k=1}^{K} \mu_{ik} \right)$. The sequencing depth $n_i$ was randomly sampled from a uniform distribution $U[6000, 17,000]$.

### Generation of simulated data using SRTsim

Our simulation strategy was inspired by the original SRTsim paper, and we used the reference-free option of SRTsim in order to strictly define genes with or without spatial variation. We used the same human cortex data as used in the SRTsim paper to define the spatial locations of 3611 spots. The tissue layer labels (six neocortical layers, L1 to L6, and the white matter) of these spots were used as categorical covariates. We used SRTsim to generate six simulated datasets, each with 1000 genes. For each dataset, we assumed 850 genes with no spatial or covariates effects ("non-SVGs") and 150 genes with constant covariate effects on L1 to L6 ("SVGs"). In the SRTsim model, following the SRTsim paper, we set the dispersion parameter for all genes to be 0.3, and the mean parameter for the non-SVGs genes to be 0.03. For the SVGs, their mean parameter for spots outside the designated layer of focus was 0.03, and their mean parameter for spots inside the designated layer was 0.03 multiply by 3 or 1/3. In summary, the six simulated datasets represented the cases when the SVGs had higher or lower expression in layer L1 to L6 compared with other spots.

### Real data analysis

For the human cortex dataset, the layer annotations were generated by [21]. Since the layer annotations are categorical variables, we coded them as dummy variables to be used as the spot-level covariates in spVC and alternative methods. In the permutation analysis, we randomly permuted the indices of the observed spots while retaining the read counts and layer annotations as available in the real data. For every considered method, the $P$ values were corrected by the BH approach, and the type I error was calculated as the number of falsely identified genes divided by the total number of tested genes. We performed hierarchical clustering on the 2819 genes that had significant residual spatial effects by spVC and were detected in more than 500 spots. For each gene, the residual spatial effects, $\hat{\gamma}_0(\cdot)$, were calculated across the spatial spots, and then transformed using the min-max normalization. The normalized residual spatial effects were used to calculate the Euclidean distances between genes, followed by the hierarchical

clustering. The Gene Ontology (GO) enrichment analysis was performed using the ClusterProfiler [46] package. Significant GO terms (restricted to biological processes) were identified using a threshold of 0.05 on BH-adjusted *P* values.

For the mouse cerebellum dataset, the cell type weights were estimated by the RCTD method as described in Cable et al. [18] and then normalized to obtain cell type proportions. In our analysis, we only used the proportions of the top six major cell types (granule, oligodendrocytes, astrocytes, MLIs, Bergmann, and Purkinje), which had an overall proportion of 80.8% on average. The proportions of these six cell types were treated as spot-level covariates in spVC and alternative methods. In the permutation analysis, we randomly permuted the indices of the observed spots while retaining the read counts and cell type proportions as in the real data. For every considered method, the *P* values were corrected by the BH approach, and the type I error was calculated as the number of falsely identified genes divided by the total number of tested genes. For application of spVC to the real data, in addition to hypothesis testing, we also calculated the deviance measure to quantitatively evaluate the contributions of specific components to explain the observed data. To investigate the contribution of the residual spatial effect, the deviance of model [4] was calculated as twice of the difference in the log-likelihood function between the saturated model and model [4]. In addition, we also calculated the deviance of a reduced model with only constant coefficients:

$$g\{\mu(\boldsymbol{x}_i, \boldsymbol{s}_i)\} = \log \ell_i + \beta_0 + \sum_{j=1}^{p} x_{ij}\beta_j .$$

Then, we compared the two deviance measures for the genes. To investigate the contribution of the spatially varying effects of cell type proportions, we calculated and compared the deviance of model [1] and model [4]. The GO enrichment analysis was performed as described above.

For the mouse testis dataset, we filtered out genes detected in fewer than 100 spots and spots with smaller than 500 counts. The pseudotime values of the retained spots were obtained from the original publication [31] and treated as a continuous spot-level covariate. The clustering analyses were performed using the Seurat tool [47]. To perform clustering of spots based on the estimated spatially varying effects, we first calculated the first ten principal components, and then used the FindNeighbors and FindClusters functions (resolution = 0.5) in Seurat to obtain the final clusters. To perform clustering based on the observed gene expression levels, we directly followed the Seurat pipeline starting from the count matrix, also setting resolution = 0.5.

## Supplementary Information

**Additional file 1.** Supplementary Figures S1-S24 and Tables S1-S4.

**Additional file 2.** Supplementary methods.

**Additional file 3.** Review history.

### Acknowledgements
This work was partially supported by the National Institutes of Health (NIH) NIGMS R35GM142702 (to WVL). The authors acknowledge UCR's HPC Center (HPCC), UVA's Research Computing, and NSF-MRI grant 2215705 for the computing resources made available for conducting the research reported in this paper.

**Review history**

The review history is available as Additional file 3.

**Peer review information**

**Authors' contributions**

WL and SY conceived the ideas, developed the methods, performed the simulation studies, and drafted the manuscript. SY implemented the software. WL performed the real data studies. Both authors read and approved the final manuscript.

**Availability of data and materials**

The spVC method has been implemented as an R package, which is available from https://github.com/shanyu-stat/spVC [48] The source code for reproducing the results presented in the article is available at Zenodo (DOI: 10.5281/zenodo.10946411) [49]. Both resources are released under the GNU General Public License version 3 license.
 The SpatialDE method was implemented through the Python package SpatialDE v1.1.0. The covariates were regressed out using the function `NaiveDE.regress_out` before the identification of SVGs. The SPARK and SPARK-X methods were implemented through the R package SPARK v1.1.1. For SPARK, the covariate information was provided through the `covariates` parameter, and the `fit.model` parameter was set to "gaussian". For SPARK-X, the covariate information was provided through the `X_in` parameter. The MERINGUE method was implemented through the R package MERINGUE v1.0. To account for cell/spot-level covariates, the covariates were regressed out (using linear regression) from the normalized counts obtained by function `normalizeCounts`, before the identification of SVGs.
 The human dorsolateral prefrontal cortex data [21] (Sample ID: 151673) was downloaded from https://github.com/LieberInstitute/spatialLIBD. The mouse cerebellum data [18] was downloaded from at https://singlecell.broadinstitute.org/single_cell/study/SCP948. The mouse testis data [31] was downloaded from https://www.dropbox.com/s/ygzpj0d0oh67br0/Testis_Slideseq_Data.zip?dl=0.

## Declarations

**Ethical approval and consent to participate**
Not applicable.

**Consent for publication**
Not applicable.

**Competing interests**
The authors declare that they have no competing interests.

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

## 