## [**Additional file 3.** Review history. · Genome Biology]

Review History

First round of review

Reviewer 1

Were you able to assess all statistics in the manuscript, including the appropriateness of statistical tests used? Yes.

Were you able to directly test the methods? Yes.

Comments to author:

In this manuscript, Yu et al. present spVC, a method to discover spatially variable genes based on a generalized Poisson model. They validated the rationality and effectiveness of their method with the synthetic data with known underlying expression patterns. They compared their method with another four methods developed for SVG detection and found that their method has significant advantages. As spatial transcriptomic data become more widespread, mathematical modeling and data analysis become important computational challenges. I believe this paper addresses a relevant and timely issue. We have the following specific comments:

1. spVC models the gene expression in every spot with a spatial conditional Poisson model and decomposes the expected gene expression into four components: the intercept, the spatially invariant effect of the covariates, the spatially varying effects of the covariates, and the unexplained residual spatial effects. The authors show the estimated residual spatial effects (γ_0) and estimated spatially varying effects (γ_i) in different examples. Can they show all four components? Especially, They can compare spVC with existing methods to reveal which components have been characterized by existing algorithms.
2. In the generation of simulated data section, the authors model the sequencing process of ST-seq and scRNA-seq with a multinomial distribution and the gene expression process with a spatial conditional Poisson process. In this modeling framework, how to get a Poisson regression model with a link function in Eq. (1). The authors can refer article (Curr. Bioinform. 2023, DOI: 10.2174/1574893618666230529145130) and give a more detailed explanation.
3. How to randomly permute the data so that none of the genes have any residual spatial effects? How to compute the type I error for different algorithms? Please clearly provide the method.
4. In the model estimation section on page 17: more details about the variance function and quasi-likelihood function are very beneficial for readers.
5. I suggest the author provide three detailed algorithm flowcharts: model estimation modular, hypothesis testing modular, and simulation modular.

Reviewer 2

Were you able to assess all statistics in the manuscript, including the appropriateness of statistical tests used? Yes.

Were you able to directly test the methods? No.

Title: spVC for the detection and interpretation of spatial gene expression variation

Manuscript ID: GBIO-D-23-01384

This paper proposes a new statistical method to detect spatially variable genes (SVGs). The method uses a generalized Poisson model to fit the gene expressions on other covariates and functional spatial coordinates. The statistical method looks solid. However, the connection with biological implications is weak and vague.

Major concerns:

1. The paper provides statistical interpretations of the generalized Poisson model (1). However, it is unclear how to biologically interpret $\gamma_0(s)$ and $\gamma_j(s)$?
 - (a) Why separating spatially varying covariate effect and residual spatial effect is important? What additional biological insights can we gain from the model? What would be next steps to investigate the biological implications of the detected SVGs? The paper has applied the method to several experimental datasets, identified some SVGS, and did GO studies. However, the explanations of the analysis results were superficial, and makes these analyses look like superficial demonstrations of a statistical method.
 - (b) What does $\gamma_j(s)$ mean biologically when the j th covariate is cell type proportion, cell type indicator, tissue structure indicator, respectively? What biological insights can we obtain under these different cases?
2. The authors need to provide more evidence to support fitness of the generalized Poisson model in the experimental data. I like the idea that the author permute the spatial coordinates to check the type I error. However, I think the permutation should be performed many times to check the null distributions of T_j^α in real data. Is the realized null distribution similar to the theoretical distributions? Also, is α a tuning parameter? If so, how to choose it?
3. The authors mentioned three types of hypotheses in the methods.
 - (a) On page 19 line 16, the authors mentioned that “We only consider the spatially varying effect of a covariate if the constant effect of this variable is significant and the gene demonstrates significant residual spatial patterns in the reduced model.” Why does this procedure make sense from a biological perspective? A gene cannot have cell-type-specific spatial patterns if they do not show global spatial patterns?
 - (b) The authors did not clarify which one was tested in each dataset. For example, on page 8 line 54, the authors mentioned that “spVC had a type I error of 0.001”. Which hypothesis does this type I error correspond to? Similar problems exist in other dataset analysis. When compared to other methods, did all the methods test the same hypothesis?
4. For simulation studies, the authors simulated the data according to model (1). It is unclear how the methods will perform if the simulated data do not exactly follow the

proposed model. In recent years, many spatial transcriptomics data simulators have been proposed such as SRTsim [Zhu et al., 2023] and scDesign3 [Song et al., 2023]. These simulators have certain advantages to generate more realistic spatial transcriptomics data. How does spVC perform on the data simulated by those methods?

5. What are the recommended pre-analysis and down-stream analysis for spVC? For example, spVC can be used on data with pre-derived cell type labels, or cell type proportions. What are the recommended methods to derive cell type labels or cell type proportions? Can the clustering or decomposition method utilize spatial information? Will this cause double-dipping problem and cause inflated type I error? For the downstream analysis, what further analyses can be done other than GO analysis and why?
6. The authors need to report the computation cost of the method, including the memory usage and computation time.

Minor concerns:

1. The authors need to clarify how the competitive methods were applied. For example, on page 22 line 14, when applying StpatialDE, the authors mentioned that “The covariates were regressed out using the function `NaiveDE.regress_out` before the identification of SVGs”. What are those covariates? I have the same question for SPARK and SPARKX.
2. On page 19 line 16, “cvoariate” should be “covariate”.

References

- Dongyuan Song, Qingyang Wang, Guanao Yan, Tianyang Liu, Tianyi Sun, and Jingyi Jessica Li. scdesign3 generates realistic in silico data for multimodal single-cell and spatial omics. *Nat Biotechnol*, May 2023. doi: 10.1038/s41587-023-01772-1.
- Jiaqiang Zhu, Lulu Shang, and Xiang Zhou. Srtsim: spatial pattern preserving simulations for spatially resolved transcriptomics. *Genome Biol*, 24(1):39, Mar 2023. doi: 10.1186/s13059-023-02879-z.

Response to Reviewers' Comments

We deeply appreciate the constructive comments and suggestions provided by the editor and reviewers. We have revised our manuscript based on these comments and suggestions. Before we provide a point-by-point response to both reviewers, we briefly summarize the major revisions in response to these invaluable inputs below.

- **Comprehensive evaluation on simulated data.** We have augmented the evaluation of spVC on simulated data, showcasing its ability to accurately discern the directions of association between gene expression and covariates, as well as its precision in estimating covariates' and residual spatial effects across genes.
- **New real data application.** We conducted a new real data application to a mouse testis dataset to further demonstrate the advantage of spatial gene expression variation decomposition. This analysis demonstrates the potential of spVC to identify spot clusters representing different stages of the seminiferous epithelium cycle by estimating the spatially varying effects of spot pseudotimes and characterizing the spatio-temporal dynamics of gene expression during spermatogenesis. This potential is obscured when analyzing observed gene expression alone.
- **Assumption evaluation of spVC.** We further evaluated the assumptions of spVC using the permutation analyses, confirming that the realized null distributions of the test statistics are similar to the theoretical distributions.
- **New simulator validation of spVC.** We have considered another simulator SRTsim which allows for the simulation of genes with constant covariate effects. Using data simulated by SRTsim, we obtained results consistent with our own simulation studies, supporting the accuracy of spVC.
- **New downstream and comparison analyses.** We performed a survey of typical downstream analyses performed in SVG studies and demonstrated in the revised manuscript that spVC is not only compatible with these typical analyses, but also enables more detailed analysis and categorization of SVGs. Newly added analyses during the revision include comparisons between spVC-identified genes with previously reported cortex marker genes or cell-type-associated genes (human cortex and mouse cerebellum datasets), comparisons between genes identified by spVC and alternative methods (human cortex and mouse cerebellum datasets), validation of spVC-identified genes using smFISH images (mouse testis dataset), and identification of spot clusters based on spatially varying covariate effects (mouse testis dataset).
- **Correction and clarity enhancement.** During the revision, we identified an issue with the previous analysis script for the human cortex dataset. We have corrected the issue and updated all relevant results in the revised manuscript. We have also revised the manuscript and added illustration figures to improve the clarity of the simulated data generation process, the estimation of spVC, and the real data analyses.
- **Package update.** The spVC package has been updated to enhance user-friendliness based on the comments received.

Reviewer #1:

In this manuscript, Yu et al. present spVC, a method to discover spatially variable genes based on a generalized Poisson model. They validated the rationality and effectiveness of their method with the synthetic data with known underlying expression patterns. They compared their method with another four methods developed for SVG detection and found that their method has significant advantages. As spatial transcriptomic data become more widespread, mathematical modeling and data analysis become important computational challenges. I believe this paper addresses a relevant and timely issue. We have the following specific comments.

Response:

We appreciate the reviewer's thoughtful comments and acknowledgment of the relevance and timeliness of our manuscript. Additionally, we are grateful for the recognition of the validation of our method using synthetic data. We have carefully considered the reviewer's comments and have incorporated revisions into our manuscript. The updated contents are highlighted in red for clarity. Please see our point-by-point responses below.

1. spVC models the gene expression in every spot with a spatial conditional Poisson model and decomposes the expected gene expression into four components: the intercept, the spatially invariant effect of the covariates, the spatially varying effects of the covariates, and the unexplained residual spatial effects. The authors show the estimated residual spatial effects (γ_0) and estimated spatially varying effects (γ_i) in different examples. Can they show all four components? Especially, They can compare spVC with existing methods to reveal which components have been characterized by existing algorithms.

Response:

We appreciate the reviewer's suggestions on the evaluation in our simulation analysis. In our original manuscript, we compared spVC with alternative methods based on their type I errors and statistical power, but not the estimation accuracy of the individual components. This was because spVC is the first method that tries to decompose the expected gene expression into an intercept, constant effects of covariates, spatially varying effects of the covariates, and unexplained residual spatial effects. Even when alternative methods can account for the constant effects of covariates, they only output P values of the spatial gene expression variation, but not the individual components. We provide a more detailed description of each method below:

- SpatialDE models spatial variation through a spatial covariance matrix in a multivariate normal model. It does not estimate spatial effect as a spatial function on the provided domain. When spot-level covariates are available, constant covariate effects are regressed out but not outputted by the software.
- SPARK models spatial effect as a Gaussian process, which is treated as an additive term in the link function of a Poisson distribution. When spot-level

covariates are available, constant covariate effects are accounted for in the link function but not outputted by the software.

- SPARK-X uses a non-parametric approach to identify significant spatial variation, and spatial effects are not directly modeled. It can also account for constant covariate effects, which are not outputted by the software.
- MERINGUE uses the spatial autocorrelation measure, Moran's I, and its statistical significance to quantify the association between gene expression and spatial locations. The method does not have an internal step to model spot-level covariates, but we performed a linear regression to remove constant covariate effects before carrying out its test.

In our original manuscript, we demonstrated the estimated spatially varying effects and residual spatial effects using a few example genes, and we agree with the reviewer that a more comprehensive evaluation would be helpful. Therefore, in the revised manuscript, in addition to the individual gene examples, we compared the complete sets of true and estimated values for different components. This encompassed intercepts (for all 4 groups; Supplementary Figure S2), constant effects of the covariates (for Groups 2 to 4; Supplementary Figure S3), spatially varying effects of the covariates (for Group 4; Supplementary Figure S4), and unexplained residual spatial effects (for Groups 3 and 4; Supplementary Figure S5). The results provided further validation of spVC's capacity to accurately discern the directions of association between gene expression and covariates, as well as its precision in estimating covariates' and residual spatial effects across genes. Notably, in Group 4, as spot size increased from 500 to 8000, the median Pearson correlation between true and estimated residual spatial effects (evaluated at the observed spots) rose from 0.75 to 0.95 (Figure **R1A-B** in this response letter). Similarly, the median correlation between true and estimated covariates' spatial effects increased from 0.89 to 0.96 (Figure **R1C**). These findings underscore the importance of sample size in spatial variation analysis of sparse transcriptomics data.

Figure R1. Evaluation on simulated data. **A:** Comparison between estimated and true residual spatial effects for genes in Group 3. For each gene, the true residual spatial function and the estimated residual spatial function were both evaluated at the observed spots, and the Pearson correlation between the two was calculated. **B:** Comparison between estimated and true residual spatial effects for genes in Group 4. **C:** Comparison between estimated and true spatial effects of covariates for genes in Group 4. For each gene, the true spatial effects function and the estimated spatial function were both evaluated at the observed spots, and the Pearson correlation between the two was calculated.

We have added the new analysis and results to pages 6, 8 and Supplementary Figures S2-S5 of the revised manuscript.

2. In the generation of simulated data section, the authors model the sequencing process of ST-seq and scRNA-seq with a multinomial distribution and the gene expression process with a spatial conditional Poisson process. In this modeling framework, how to get a Poisson regression model with a link function in Eq. (1). The authors can refer article (Curr. Bioinform. 2023, DOI: 10.2174/1574893618666230529145130) and give a more detailed explanation.

Response:

We appreciate the reviewer’s suggestion regarding our “Generation of simulated data” section. We would like to clarify that while there are similarities between the estimation model used in spVC and the generation model of the simulated data, they are not identical.

In the spVC model, we assume an independent Poisson model for each gene, where the log-transformed mean is decomposed into the intercept, constant effect of covariates, spatially varying effects of covariates, and the residual spatial effects, as specified in formula (1). In the generation of simulated data, the true expression levels of genes are simulated using the log link functions. The specific components in the link functions depend on which of the four gene groups a gene belongs to. Subsequently, a multinomial distribution is used to model the sequencing process and generate simulated read counts based on the true gene expression levels. In the revised manuscript, we have referenced the review article and revised the “Generation of simulated data” section to improve its clarity (pages 20 and 23-25).

3. How to randomly permute the data so that none of the genes have any residual spatial effects? How to compute the type I error for different algorithms? Please clearly provide the method.

Response:

We apologize for any confusion caused. We carried out permutation analyses on the human cortex dataset and the mouse cerebellum dataset. In each analysis, we randomly permuted the indices of the observed spots while retaining the read counts and covariate information as observed in the real data. Consequently, none of the genes were expected to have any significant residual spatial effects, and any genes identified by a computational method should be considered as false positives. Since all considered methods can provide P values for the residual spatial effects, for each method, we first performed multiple testing correction using the Benjamini-Hochberg approach, and then calculated type I errors as the number of falsely identified genes divided by the total number of tested genes. The main reason for focusing on testing the residual spatial effects in permutation analysis was that alternative methods are not able to account for spatially varying effects of covariates. We have clarified the above points in the “Results” section (page 8) and the “Real data analysis” section (page 25) of the revised manuscript.

4. In the model estimation section on page 17: more details about the variance function and quasi-likelihood function are very beneficial for readers.

Response:

We thank the reviewer for this suggestion. In the revised manuscript (page 21), we have provided a more detailed description about the definitions of the variance function and the quasi-likelihood function. We have also added a reference for the general quasi-likelihood approach.

5. I suggest the author provide three detailed algorithm flowcharts: model estimation modular, hypothesis testing modular, and simulation modular.

Response:

We appreciate these suggestions. In the revised manuscript, we have added two illustration figures, Figure **R2** for the model estimation and testing procedures and Figure **R3** for the simulation procedure. We display both figures below and have added them as Supplementary Figures S22 and S25 of the revised manuscript.

Figure R2. Overview of the spVC method. **A:** The required input of spVC includes the spatial transcriptomics data (the read count matrix and the corresponding spatial location matrix) and the spot-level covariate data. The covariates should be provided for the same spots observed in the spatial transcriptomics data. **B:** The four main steps in spVC's estimation procedure. **C:** The two-step testing procedure used in our article. **D:** For each gene, spVC outputs the estimated constant and spatial effects as well as their corresponding P values.

Figure R3. Generation process of the simulated data. For a given spot number l , we first generated the two-dimensional spatial coordinates by randomly selecting l data points from the observed locations of the mouse cerebellum data. Next, we generated the expected expression levels of four groups of genes. Each group contained 5000 genes. We assumed that genes in Group 1 only depended on an intercept term; genes in Group 2 depended on an intercept and constant covariate effects; genes in Group 3 depended on an intercept, constant covariate effects, and residual spatial effects; genes in Group 4 depended on an intercept, constant covariate effects, spatially varying covariate effects, and residual spatial effects. After we obtained the expected expression levels of all genes across the l spots, the final read counts were simulated based on a multinomial sampling process for each spot.

Reviewer #2:

This paper proposes a new statistical method to detect spatially variable genes (SVGs). The method uses a generalized Poisson model to fit the gene expressions on other covariates and functional spatial coordinates. The statistical method looks solid. However, the connection with biological implications is weak and vague.

Response:

We appreciate the thoughtful consideration of our manuscript by the reviewer and their assessment of the solidity of our statistical approach. We acknowledge the concern regarding the connection with biological implications. In our revisions, we have endeavored to strengthen the link between our statistical methodology and its biological implications. We have provided additional analyses and explanations to illustrate how spVC can contribute to our understanding of spatial gene expression patterns in biological systems. The updated contents are highlighted in red in the revised manuscript for clarity. Please see our point-by-point responses below.

Major concerns:

1. The paper provides statistical interpretations of the generalized Poisson model (1). However, it is unclear how to biologically interpret $\gamma_0(s)$ and $\gamma_j(s)$?

(a) Why separating spatially varying covariate effect and residual spatial effect is important? What additional biological insights can we gain from the model? What would be next steps to investigate the biological implications of the detected SVGs? The paper has applied the method to several experimental datasets, identified some SVGs, and did GO studies. However, the explanations of the analysis results were superficial, and makes these analyses look like superficial demonstrations of a statistical method.

Response:

We acknowledge the reviewer's questions regarding the interpretations of the identified spatial effects by spVC. In response, we have carefully revised and strengthened the manuscript to address these questions.

We appreciate the reviewer's inquiry regarding the significance of separating spatially varying covariate effects ($\gamma_j(s)$) and residual spatial effects ($\gamma_0(s)$). First, by isolating the spatially varying effects ($\gamma_j(s)$) of spot-level covariates, researchers can discern how these factors influence gene expression patterns across different spatial locations. This helps uncover the biological mechanisms underlying spatial gene expression regulation and provides insights into cellular heterogeneity and tissue organization. In addition, spatially varying effects ($\gamma_j(s)$) of spot-level covariates can also suggest interactions between observed and unobserved spot-level factors. Second, estimation of residual spatial effects ($\gamma_0(s)$) enables the identification of spatial patterns in gene expression that may not be directly attributable to the measured covariates. This exploration can lead to

the discovery of spatially regulated genes that respond to unmeasured factors, such as local microenvironments or cellular interactions, contributing to our understanding of tissue function and development.

In fact, the concept of covariates' spatially varying effects is not new in biostatistics, and they have been used to in diverse applications such as spatial disease modeling¹, neuroimaging², and ecology³. However, such a tool to decompose spatial gene expression variation which is tailored for spatial transcriptomics data has been lacking. As the accumulation of new spatial transcriptomics data accelerates, understanding the spatial variation of gene expression becomes increasingly pivotal in formulating biological hypotheses. We believe that the development of spVC addresses a significant gap in this area. It's important to note that while identifying the spatially varying effects and residual spatial effects and evaluating their statistical significance is a major contribution of spVC, it's not the sole one. Equally significant is spVC's ability to estimate continuous spatial functions that characterize the spatial changes of these effects. While some alternative methods compared in our manuscript can account for constant covariate effects, they lack the capability to provide such detailed estimation.

In the original manuscript, we used tissue layers (categorical) or cell type proportions (continuous) as the covariates in the spVC model, since gene expression heterogeneity in different tissue layers or cell types is a key question in single-cell genomics, and these two types of information are relatively easier to obtain in real data applications. To further demonstrate the potential advantage of decomposition of spatial gene expression variation, we conducted another real data application in the revised manuscript. We applied spVC to a mouse testis dataset obtained using the Slide-seq technology. After filtering, this dataset comprised 10,527 genes and 18,097 spots. In order to analyze the spatio-temporal gene expression regulation, we used the pseudotime orders inferred by Chen et al. (authors of the testis dataset) as a continuous covariate in this application. These pseudotime orders confirmed the recognized developmental trajectory of germ cells, which starts from the basement membrane and progresses towards the lumen of the seminiferous tubules (Figure **R4A** in this response letter).

We first identified 8998 genes with a significant constant coefficient of the pseudotime orders. Among these, 1370 genes exhibited increased expression in the trajectory of germ cell development, while the remaining 7628 genes showed decreased expression. For example, the *Smcp* gene, which encodes a sperm mitochondria-associated cysteine-rich protein, had a positive coefficient of 3.79 (adjusted $P \approx 0$), indicating higher expression in the later stage of the germ cell development trajectory as confirmed by single-molecule fluorescence *in situ* hybridization (smFISH) image (Figure **R4B**). Conversely, the *Lyar* gene, which encodes a cell growth-regulating nucleolar protein, had

¹ Osei FB, Stein A, Andreo V. A zero-inflated mixture spatially varying coefficient modeling of cholera incidences. *Spatial statistics*. 2022 Apr 1;48:100635.

² Zhu H, Fan J, Kong L. Spatially varying coefficient model for neuroimaging data with jump discontinuities[J]. *Journal of the American Statistical Association*, 2014, 109(507): 1084-1098.

³ Finley A O. Comparing spatially-varying coefficients models for analysis of ecological data with non-stationary and anisotropic residual dependence[J]. *Methods in ecology and evolution*, 2011, 2(2): 143-154.

a negative coefficient of -2.91 (adjusted $P \approx 0$), and was shown to have higher expression in the earlier stage of the development trajectory by smFISH image (Figure R4B). Subsequently, we identified 546 genes that presented a significant spatially varying coefficient of the pseudotime orders, suggesting a dynamic interplay between spatial and temporal dimensions. To further understand these identified spatial effects, we clustered the spots based on their corresponding spatially varying coefficients ($\gamma_1(s)$) of the 546 genes, and obtained five spot clusters (Figure R4C; Methods). The median pseudotime of these clusters was similar, indicating that the clustering analysis based on spVC's estimated spatial coefficients was not confounded by the actual pseudotime values of the spots (Figure R4D). In contrast, direct clustering analysis using observed gene expression yielded eight spot clusters with apparent differences in their pseudotime values, suggesting that the clustering process was primarily driven by expression changes along the germ cell development trajectory (Figure R4E-F).

Figure R4. Application of spVC to the mouse testis dataset. **A:** Pseudotime values of the observed spatial spots. **B:** Relative expression levels of *Lyar* and *Smcp*. The read counts were normalized by library size, log-transformed, and then scaled by the min-max normalization to obtain the relative expression levels. **C:** Five spot clusters identified based on the spatially varying effects ($\gamma_1(s)$) of pseudotime inferred by spVC. **D:** Distribution of pseudotime values within the five clusters shown in C. **E:** Eight spot clusters identified based on the observed gene expression levels. **F:** Distribution of pseudotime values within the eight clusters shown in E. **G:** Relative expression levels of *Prm2* and *Tnp1* within the five clusters shown in C. **H:** Estimated spatially varying effects of *Prm2* and *Tnp1* within the five clusters shown in C.

We further scrutinized the five spot clusters unveiled by spVC and observed significant differences in genes previously known to be associated with various stages of the

epithelium cycle of seminiferous tubules, including *Prm1* ($P \approx 0$), *Tnp1* (4.68×10^{-192}), *Prm2* (4.57×10^{-122}), *H3f3b* (5.11×10^{-64}), *Habp4* (1.84×10^{-18}), *Trim24* (4.68×10^{-4}), *Smarca2* (6.54×10^{-4}), and *Ezh2* (9.30×10^{-4}). The P values were obtained by carrying out the Kruskal-Wallis test on relative gene expression levels followed by the BH correction. For example, *Prm1* was shown to be associated with step 15 spermatids onward and *Tnp1* was shown to be associated with spermatids of steps 9–12. In addition to presenting significant difference in observed gene expression, these two genes demonstrated more obvious cluster-dependent patterns in the spatial coefficients estimated by spVC (Figure **R4G-H**). The germ cell development cycle and the seminiferous epithelium cycle are two interrelated yet distinct processes occurring within the testes, working together to ensure a continuous production of spermatozoa. Our analysis demonstrates the potential of spVC to unveil spot clusters representing different stages of the seminiferous epithelium cycle by estimating the spatially varying effects of spot pseudotimes and characterizing the spatio-temporal dynamics of gene expression during spermatogenesis. This potential is obscured when analyzing observed gene expression alone. We have added descriptions of this new analysis to pages 17 and 18 of the revised manuscript.

We understand that the reviewer has also raised a question about the downstream analysis following the application of spVC, which is closely related to Comment #5 of the reviewer. Therefore, we have provided a complete response under Comment #5.

(b) What does $\gamma_j(s)$ mean biologically when the j th covariate is cell type proportion, cell type indicator, tissue structure indicator, respectively? What biological insights can we obtain under these different cases?

Response:

The primary objective of our work is to introduce a statistical model and corresponding estimation and inference tools to decompose expected gene expression into four components: intercept, constant effects of the covariates, spatially varying effects of the covariates, and unexplained residual spatial effects. The interpretations of the spatially varying effects are contingent upon the specific covariates utilized in each application. In general, significant spatially varying effects of a covariate imply that this factor exerts varying degrees of influence on gene expression across different spatial locations. It suggests the presence of unobserved factors that interact with the observed covariate in the regulation of gene expression.

In the current manuscript, we illustrate the application of spVC using three real datasets. In the analysis of the mouse cerebellum data, cell type proportions served as the spot-level covariates. When a gene exhibits only a significant constant covariate effect, it implies that the gene's expression can be predicted based on cell type proportions alone. Conversely, if a gene demonstrates a significant spatially varying covariate effect, it suggests the presence of additional external factors (e.g., local microenvironments or cellular interactions) that interact with cell type identities to impact gene expression. In our manuscript, we highlight the cases of three example genes, *Aldoc*, *Calb2*, and

Snhg11, which exhibit significant spatially varying effects of cell type proportions (Figure 7D-G). We have displayed these panels as Figure R5 below for the reviewer's convenience. These genes play crucial biological roles in the cerebellum, and some have been reported to display heterogeneous expression within cell types. With the application of spVC, we quantified their spatial heterogeneity within cell types using the spatially varying coefficients. We anticipate that spVC will serve as a valuable tool to assist researchers in categorizing genes based on their spatial heterogeneity within cell types.

Figure R5. Application of spVC to the mouse cerebellum data. **A:** Relative expression levels of *Aldoc*, *Calb2*, and *Snhg11*. The read counts were normalized by library size, log-transformed, and then scaled by the min-max normalization to obtain the relative expression levels. **B:** Estimated spatially varying effects of cell type proportions for *Aldoc* (Purkinje cells), *Calb2* (granule cells), and *Snhg11* (granule cells). **C:** Expected gene expression of *Aldoc*, *Calb2*, and *Snhg11* estimated by the reduced model which did not consider the spatially varying coefficients of cell type proportions. The shown values were scaled by the min-max normalization. **D:** Expected gene expression of *Aldoc*, *Calb2*, and *Snhg11* estimated by the full spVC model. The shown values were scaled by the min-max normalization.

In the newly added analysis of mouse testis data (described in our response to Comment #1a), pseudotime orders of sperm cells serve as spot-level covariates. In this context, spatially varying effects of pseudotime orders represent spatio-temporal interplays in the regulation of gene expression levels. Estimating the spatially varying effects allows us to quantify how each gene's expression is regulated in a spatially and temporally coordinated manner, which serves as a key step to unravel the complexities of developmental process. Through our analysis, we showed that the developmental cycle of seminiferous tubules can be unveiled by identifying spatio-temporal interactions using spVC, but is obscured by analyzing the observed data alone.

In the analysis of the human cortex data, we utilized the tissue structure indicators as the spot-level covariates. We didn't consider the spatially varying effects of covariates in this particular analysis, since the tissue layer indicators exhibited clustered patterns rather than dispersed patterns. In this case, it was methodologically infeasible to distinguish between the spatially varying effects and the residual spatial effects, as explained in the Methods section. Yet, in other scenarios where it is feasible to estimate the spatially

varying effects of tissue structure or cell type indicators, their presence indicates that the association between gene expression and tissue structure or cell type changes across spatial locations. Like in the analysis of cell type proportions, such phenomenon suggests the interaction between tissue structure or cell type identities with unmeasured factors such as local microenvironments.

Lastly, as spatially resolved transcriptomics technologies continue to be applied to study a wide range of diseases, we anticipate that spVC can be utilized to account for disease-related covariates. This application has the potential to offer valuable insights into the dysregulation of gene expression across diverse disease conditions.

2. The authors need to provide more evidence to support fitness of the generalized Poisson model in the experimental data. I like the idea that the author permute the spatial coordinates to check the type I error. However, I think the permutation should be performed many times to check the null distributions of T_j^α in real data. Is the realized null distribution similar to the theoretical distributions? Also, is α a tuning parameter? If so, how to choose it?

Response:

We thank the reviewer for this suggestion and appreciate the idea to check the null distributions using permutation analysis. We have added this additional step to our permutation analysis during the revision. In this manuscript, we follow Wood (2013)⁴ to construct the null distribution of the test statistics, whose degrees of freedom depend on several smooth parameters. As the smooth parameters are selected by generalized cross validation and are data-dependent, the null distributions of the same gene vary between different permutation replicates. Therefore, it is not feasible to directly compare the realized null distributions and the theoretical distributions. Instead, for each gene, we compared the P values obtained from the permutation replicates and the uniform distribution, which is the theoretical distribution under the null hypothesis. For both the human cortex and mouse cerebellum datasets, we repeated the permutation procedure 100 times and obtained 100 P values for each gene. Figure **R6A-B** illustrates the observed and expected quantiles of P values for two example genes (*GNB1* and *GPR88*) in the human cortex data, and Figure **R6C-D** illustrates two example genes (*Glr3* and *Kif1a*) in the mouse cerebellum data. To quantitatively evaluate the results across all genes, we calculated the root of mean squared error (RMSE) between the observed quantile-quantile curve and the $y = x$ reference line for each gene. On average, the RMSE on the mouse cerebellum data was 0.014, and the RMSE on the human cortex data was 0.026. These results indicate that the theoretical distributions are appropriate for the test statistics. We have added the above analysis and results to pages 8, 10, and 12 of the revised manuscript.

⁴ Wood S N. On p-values for smooth components of an extended generalized additive model. *Biometrika*, 2013, 100(1): 221-228.

Figure R6. A-B: Observed quantiles of P values vs. expected quantiles (based on the uniform distribution) for genes *GNB1* and *GPR88* in the human cortex data. **C-D:** Observed quantiles of P values vs. expected quantiles (based on the uniform distribution) for genes *Glr* and *Kif1a* in the mouse cerebellum data.

Regarding the parameter α , it must be chosen in a data-dependent manner. In our manuscript, we adopt the method of a well-behaved Wald statistic described in Section 2.2 of Wood (2013)⁵. On page 23 of the revised manuscript, we have provided a more detailed description of how α is calculated.

3. The authors mentioned three types of hypotheses in the methods.

(a) On page 19 line 16, the authors mentioned that “We only consider the spatially varying effect of a covariate if the constant effect of this variable is significant and the gene demonstrates significant residual spatial patterns in the reduced model.” Why does this procedure make sense from a biological perspective? A gene cannot have cell-type-specific spatial patterns if they do not show global spatial patterns?

Response:

We apologize for any confusion caused. In our analyses, we only tested if a gene exhibited significant spatially varying covariate effects using a full model (formula (1)) if

⁵ Wood S N. On p-values for smooth components of an extended generalized additive model. *Biometrika*, 2013, 100(1): 221-228.

the gene demonstrated significant constant covariate effects and significant residual spatial effects in the reduce model (formula (4)). We adopted this two-step testing procedure to enhance the efficiency and interpretability of the analyses, aligning with the common practice in regression modeling where interaction terms are considered only when main effects are significant. Yet, we acknowledge the possibility that a gene may lack constant covariate effects or global spatial effects while still displaying spatially varying covariate effects. Therefore, we have updated our spVC software to offer users the choice between the two-step testing procedure, which prioritizes efficiency, and the full testing approach, which provides more comprehensive results. We have elaborated on this adjustment on page 22 of the revised manuscript.

(b) The authors did not clarify which one was tested in each dataset. For example, on page 8 line 54, the authors mentioned that “spVC had a type I error of 0.001”. Which hypothesis does this type I error correspond to? Similar problems exist in other dataset analysis. When compared to other methods, did all the methods test the same hypothesis?

Response:

We thank the reviewer for this comment, and would like to address two key points. First, when multiple methods were compared in the same analysis, all methods were evaluated on the same hypothesis. Second, since spVC is the only method that can estimate and test spatially varying covariate effects, alternative methods were not evaluated on this specific task. To provide further clarity:

- In the simulation study, all methods were compared based on their type I error and statistical power for detecting the residual spatial effects in the presence of covariates. In addition, spVC' statistical power for detecting spatially varying effects of covariates was evaluated.
- In the permutation analyses of real datasets, all methods were compared based on their type I error in detecting genes' residual spatial effects in the presence of covariates.

We have also revised our manuscript to clarify the above points.

4. For simulation studies, the authors simulated the data according to model (1). It is unclear how the methods will perform if the simulated data do not exactly follow the proposed model. In recent years, many spatial transcriptomics data simulators have been proposed such as SRTsim [Zhu et al., 2023] and scDesign3 [Song et al., 2023]. These simulators have certain advantages to generate more realistic spatial transcriptomics data. How does spVC perform on the data simulated by those methods?

Response:

We thank the reviewer for the comment and suggestions. We first would like to clarify that the data generation model in our simulation is not exactly the same as the estimation model used in spVC. There are two major differences. First, spVC assumes an independent Poisson distribution for every gene, while the simulated read counts in a spot

are jointly generated for all genes using a multinomial distribution. Second, the spatial effects in the simulated data are generated using exponential or cosine functions. In contrast, in spVC, estimation of spatial functions is achieved through multivariate splines, and exponential or cosine functions are not directly used.

We thank the reviewer for suggesting alternative simulators such as scDesign3 and SRTsim, which use different approaches to improve the fidelity of the simulated data. We didn't use these simulators in our work since their focus is different from our study. Both scDesign3 and the referenced-based version of SRTsim aim to generate a count matrix of spatial transcriptomic data such that the synthetic data captures real data characteristics. Neither simulators are tailored for the task of analyzing spatial gene expression variation and do not consider spatially varying covariate effects. Another important distinction is that they don't provide ground truth information on spatially variable genes for the synthetic data.

The remaining option is the reference-free version of SRTsim, which is able to provide ground truth information on spatially variable genes. Using this option, we are able to specify genes that do not present any covariate or spatial effects (similar to genes in Group 1 in our original simulation; referred to as "noise genes" in SRTsim), and genes whose expression levels have a constant fold change in selected spatial regions (similar to genes in Group 2 in our original simulation; referred to as "signal genes" in SRTsim). However, there's no flexibility to introduce additional spatially varying coefficients of covariates or unexplained spatial effects. This means that we are only able to evaluate the methods' type I error in detecting residual spatial effects in the presence covariates, but not power. Nevertheless, we performed an additional simulation study, and used the reference-free version of SRTsim to generate simulated spatial transcriptomics data. Following the procedure and parameters described in the SRTsim paper, we simulated six datasets based on the spatial spots in a human cortex dataset. Each dataset contained 850 noise genes and 150 signal genes, and the six datasets differed in terms of where the signal genes were differentially expressed. Our results show that all methods except for SPARK-X had a type I error of 0; SPARK-X's type I error was between 0.86 and 0.90. The results are consistent with the observations in our original simulation. Most methods had an ideal performance in this simulation, and it was likely due to the fact that we were only allowed to set a constant mean and dispersion parameter (to be used in the Negative Binomial distribution) for all genes, which reduced the complexity of the simulated data. We would be delighted to incorporate these new results into the manuscript if the reviewer deems it necessary.

5. What are the recommended pre-analysis and down-stream analysis for spVC? For example, spVC can be used on data with pre-derived cell type labels, or cell type proportions. What are the recommended methods to derive cell type labels or cell type proportions? Can the clustering or decomposition method utilize spatial information? Will this cause double-dipping problem and cause inflated type I error? For the downstream analysis, what further analyses can be down other than GO analysis and why?

Response:

We thank the reviewer for raising the above important points. For the spot-level covariates, we used computationally inferred covariate information such as cell type proportions or pseudotime orders in our real data applications. As spatial transcriptomics technologies become increasingly popular across various physiological and pathological contexts, we anticipate the availability of directly measured covariate information, such as morphology or histology features. In our analyses, the cell type proportions were estimated by the RCTD method, and the pseudotime orders were estimated by the Monocle method (by authors of this dataset). These two methods have been shown to have good performance in previous benchmark studies, and do not utilize spatial information in their estimation. However, it's important to note that regardless of whether covariates are computationally inferred or directly measured, the current spVC method treats them as known information without accounting for potential estimation or measurement errors at this stage. Addressing these potential errors in downstream analysis poses an important statistical question and is a common challenge in the field. As a future direction, we will explore the extension of the spVC model to leverage recent post-selection inference tools to tackle these issues effectively. We have added this important issue in the Discussion section (page 19) of the revised manuscript.

As for the downstream analysis, we performed a survey of different types of downstream analysis performed in existing SVG studies (Table R1). As spVC is also a method designed for SVG study, previously considered downstream analysis is still applicable. For example, in the revised manuscript, we performed comparisons between identified SVGs with known marker genes (on human cortex, mouse cerebellum, and mouse testis datasets), gene ontology enrichment analysis (on human cortex and mouse cerebellum datasets), identification of gene sets with similar spatial patterns (on human cortex dataset), and identification of spatial domains with similar spatially varying covariate effects (on mouse testis dataset).

Table R1. Downstream analysis performed in SVG studies.

Method	Comparison with known marker genes	Gene ontology enrichment analysis	KEGG/Reactome enrichment analysis	Identification of gene sets with similar spatial expression patterns
SPARK	✓	✓	✓	
SpatialDE	✓	✓	✓	✓
SPARK-X	✓	✓	✓	
trendseek	✓			
MERINGUE	✓			✓
nnSVG	✓			
SOMDE	✓			

However, as a method focused on interpreting spatial gene expression variation, spVC offers researchers richer information to prioritize downstream analysis. Unlike existing methods, which often have heavy reliance on visualization to discern SVGs associated with covariates and those exhibiting additional unexplained spatial effects, spVC

addresses this challenge effectively. Visualization-based approaches suffer from two main drawbacks: they are impractical for scrutinizing a large number of genes, and they cannot differentiate contributions of individual covariates. In contrast, spVC offers a convenient and comprehensive evaluation of covariate effects and any residual spatial effects. Its statistical significance of different components can be used to categorize and select genes, and its estimated spatial effects can be easily visualized to facilitate covariate-specific interpretations. In studies where experimental validation such as RNA *in situ* hybridization or hybridization chain reaction can be performed, spVC can serve as a powerful tool for prioritizing gene selection in downstream validation processes.

We would like to summarize the additional downstream analysis and comparisons that we have added to the revised manuscript.

- For the human cortex dataset, to delve deeper into the layer-associated genes identified by spVC, we compared these genes with the cortex layer-associated genes studied in Maynard et al., which used a model-based method to identify the layers associated with 126 previously reported marker genes. Among these, 92 genes were detected in the spatial transcriptomics data. Maynard et al. reported associations with layers L1 to L6 for 4, 44, 46, 19, 47, and 48 genes, respectively. Our analysis revealed that the percentages of these genes positively associated with the layers, as identified by spVC (using WM as the baseline), were 25.0% (1), 31.8% (14), 41.3% (19), 36.8% (7), 42.6% (20), and 41.7% (20). These genes indeed demonstrated layer-associated expression patterns, and in particular, differential expression between the associated layers and the WM layer (Supplementary Figure S9). It is noteworthy that marker genes are expected to be a subset of significant genes identified by spVC, as spVC can potentially identify both layer-specific and layer-associated genes. We then evaluated the 2097 genes only reported to be layer-associated by spVC, and also observed apparent differences between the identified layers and WM (Supplementary Figure S10). Notably, only two of the six example genes (*NEFH* and *PCP4*) in Figure 4E were in the previously reported gene list. In contrast, the 67 genes solely reported to be layer-associated in Maynard et al. did not exhibit obvious differences between the neocortical layers and WM (Supplementary Figure S11). These comparisons demonstrate spVC's ability to identify known and novel tissue-layer-associated genes from spatial transcriptomics data. This additional analysis has been added to page 10 of the revised manuscript.
- On the human cortex dataset, we also added a comparison between the genes identified by different methods. The number of genes with significant residual spatial effects identified by SpatialIDE, SPARK, SPARK-X, and MERINGUE were 404, 5068, 11,055, and 496, respectively. We compared these genes and found that the 823 genes discovered by spVC were also identified by at least one other method, with 97.0% of its genes being identified by at least three methods. In contrast, only 19.0% of SPARK genes, 8.7% of SPARK-X genes, and 85.1% of MERINGUE genes were identified by at least three methods. We then investigated the 3022 genes which only had constant layer effects in spVC's results but were identified to have significant residual spatial effects by non-spVC methods, and found that they had much smaller estimated residual spatial effects compared with

the 823 genes identified by spVC (Supplementary Figure S12). In addition, we evaluated the nine genes that were identified as significant by all methods except for spVC, and they did not present obvious spatial variation that could not be explained by the layer distributions (Supplementary Figure S13). This additional analysis has been added to pages 10-11 of the revised manuscript.

- For the mouse cerebellum dataset, we compared the cell-type-associated genes identified by spVC with the differentially expressed marker genes identified from an snRNA-seq dataset of the mouse cerebellum, which has a median sequencing depth of a median transcript capture of 2862 unique molecular identifiers (UMIs) per cell. In contrast, the Slide-seq data used by spVC only has a median transcript capture of 329 UMIs per spot. Among the 3639 differentially expressed markers detected in both datasets, the numbers of genes upregulated in granule, Bergmann, oligodendrocyte, Purkinje, MLI, and astrocyte cells were 680, 837, 893, 2227, 742, and 783, respectively. In these cell types, the proportion of genes overlapping with those identified by spVC were 54.1% (368), 45.9% (384), 49.2% (439), 24.8% (553), 13.3% (99), and 51.7% (405), respectively. These overlapping genes demonstrated differential expression patterns associated with the corresponding cell type proportions (Supplementary Figure S17). We then evaluated the 2605 genes that were only identified from the snRNA-seq data and the 1676 genes that were only reported by spVC. The genes from snRNA-seq data did not present obviously upregulated expression levels in the claimed cell types (Figure **R7A** on next page), while the genes found by spVC presented expression patterns that were similar to those of the overlapping genes (Figure **R7B**), demonstrating the sensitivity of spVC in identifying cell-type-associated genes from sparse spatial transcriptomics data. This additional analysis has been added to pages 12 and 14 of the revised manuscript.

Figure R7. Relative expression levels of unique cell-type-associated genes. **A:** Genes were uniquely reported from snRNA-seq data. **B:** Genes were uniquely found by spVC. For genes associated with each cell type, the normalized expression levels of each gene were further scaled by the min-max normalization, and then for each spot, the average was taken across genes.

- On the mouse cerebellum dataset, we conducted a comparison between the spatially variable genes identified by spVC and those identified by alternative methods. After adjusting for the cell type proportions, spVC identified 304 genes with significant residual spatial effects, while SpatialDE, SPARK, SPARK-X, and MERINGUE identified 2183, 820, 3818, and 278 genes, respectively. We found that 96.4% of spVC's spatial genes were identified by at least one other method. However, the proportion of overlap for SpatialDE, SPARK, SPARK-X, and MERINGUE reduced to 44.2%, 84.8%, 34.2%, and 80.2%, respectively. We also found that a large proportion of genes reported to have significant residual spatial patterns by other methods were only found to be cell-type-associated by spVC (SpatialDE: 44.5%; SPARK: 58.5%; SPARK-X: 63.0%; MERINGUE: 46.8%; Supplementary Table S4). Among these genes, we visualized the top ones by *P* values, and didn't observe obvious spatial patterns not explained by the cell type proportions (Supplementary Figure S21). This additional analysis has been added to pages 16-17 of the revised manuscript.

- We have introduced the new analysis on the mouse testis dataset in response to Reviewer Comment #1. In this analysis, we emphasized example genes (*Smcp* and *Lyar*) which were identified by spVC and validated in previous RNA *in situ* hybridization experiments. Furthermore, to demonstrate the potential downstream analysis facilitated by spVC, we performed spot clustering using the spatio-temporal effects identified by spVC. This approach allowed us to identify spatial regions exhibiting differences in seminiferous tubule stages. It is important to note that such clustering patterns cannot be directly revealed using observed spatial transcriptomics data.

6. The authors need to report the computation cost of the method, including the memory usage and computation time.

Response:

We thank the reviewer for this suggestion. In the original manuscript, we have summarized the computation time of different methods on the simulated data (Figure R8A). The fastest method was SPARK-X, which uses a non-parametric approach. It finished running on all four datasets within 1 minute. spVC was the second fastest, and finished running on all four datasets within 1 hour. On the largest dataset with 8000 spots, SpatialDE, MERINGUE, and SPARK took 3.10, 16.60, and 27.25 hours to complete, respectively. We have added a comparison of maximum memory usage to page 8 of the revised manuscript. While spVC initially consumed more memory than the other methods for smaller datasets, its memory requirement increased at a slower rate as the number of spots reached 8000 (Figure R8B).

Figure R8. Computational time (**A**) and maximum memory (**B**) used on the simulated data. For methods that support parallel computation (SPARK, SPARK-X, and spVC), four cores were used.

Minor concerns:

1. The authors need to clarify how the competitive methods were applied. For example, on page 22 line 14, when applying SpatialDE, the authors mentioned that “The covariates were regressed out using the function NaiveDE.regress_out before the identification of SVGs”. What are those covariates? I have the same question for SPARK and SPARKX.

Response:

We apologize for the confusion. For each dataset, we used the same set of covariates in all considered methods to ensure fairness of comparison. For the human cortex data, the cortex layer annotations were coded as dummy variables, which served as the spot-level covariates. For the mouse cerebellum data, the proportions of six cell types (granule, oligodendrocytes, astrocytes, MLIs, Bergmann, and Purkinje) served as the spot-level covariates. We have added the above clarifications to the “Real data analysis” section (pages 25-26) of the revised manuscript.

2. On page 19 line 16, “cvoariate” should be “covariate”.

Response:

We thank the reviewer for pointing out the typo. We have corrected it in the revised manuscript.

Second round of review

Reviewer 1

The authors have addressed my concerns in the revised manuscript and I believe that this manuscript is indeed of value to the scientific community and therefore I recommend its publication.

Reviewer 2

The reviewers have addressed most of my previous comments. However, I still have a few additional comments which should not be too difficult to address.

1. The authors mentioned that they have performed a simulation study using the simulator SRTsim; however, I didn't find the results of this simulation study in the manuscript and the supplementary files. Also, I wonder why SRTsim are only used to generate genes with constant covariate effects. It seems to be able to generate data with spatially varying covariate effects and residual spatial effect.

2. I think Figure R2 is informative and clear. I suggest put this plot in the main manuscript, maybe in the introduction or in the early part of the results section.

3. In Figure 5B and 5C, what does “spatial-associated” mean? Do they correspond to “spatially varying covariate effect” or “residual spatial effect”?

4. For reproducibility consideration, the authors should submit all the codes to generate the analysis results mentioned in the paper (including numbers, figures, tables, etc.) to a public repository, such as GitHub. Also, the authors should provide necessary annotations to help readers to reproduce the results.

Response to Reviewers' Comments

We sincerely thank both reviewers for their careful evaluation of our revised manuscript and responses. We are grateful to Reviewer 1 for recommending our manuscript for publication. Additionally, we appreciate the new comments provided by Reviewer 2, which have contributed to the enhancement of our manuscript. Please refer to our point-by-point response below for detailed explanations. We have also included an annotated PDF version of the revised manuscript with changes highlighted in red as a supplementary file.

Reviewer #2:

The authors have addressed most of my previous comments. However, I still have a few additional comments which should not be too difficult to address.

1. The authors mentioned that they have performed a simulation study using the simulator SRTsim; however, I didn't find the results of this simulation study in the manuscript and the supplementary files. Also, I wonder why SRTsim are only used to generate genes with constant covariate effects. It seems to be able to generate data with spatially varying covariate effects and residual spatial effect.

Response:

We apologize for the confusion. As explained in our previous response, we didn't include the results based on SRTsim due to their alignment with our initial findings and the constraints of SRTsim within our evaluation frameworks. In this revised version, we have incorporated a concise overview of the results on page 9 and provided a description of the simulation methodology on pages 26-27 of the revised manuscript. Below, we provide the excerpt of the results description:

Lastly, we performed a simulation study using an independent simulator named SRTsim, which allowed us to generate spatial transcriptomics data while specifying whether a gene has constant covariate effects. We used SRTsim to generate six simulated datasets (see Methods for details). Each dataset included 150 genes with constant covariate effects (similar to Group 2 genes in our previous simulation) and 850 genes with no spatial or covariate effects (similar to Group 1 genes in our previous simulation). Then, we evaluated the type I errors of different methods in identifying residual spatial effects. Consistent with our previous simulation findings, we observed that all methods except for SPARK-X had a type I error of 0; SPARK-X's type I error was between 0.86 and 0.90. Additionally, we noted that most methods demonstrated improved type I error control in this simulation, likely attributed to the constraint of setting a constant mean and dispersion parameter (of the Negative Binomial distribution) for all genes, thereby reducing the complexity of the simulated data.

We used the Shiny application of SRTsim to generate the simulated data. Figure R1 on the next page provides a screenshot of the Shiny application. In order to specify the relationship between gene expression and a categorical covariate, users need to change setting in the "Group Assignment" panel in Figure R1. In the current setting, only a

constant value of fold change is allowed, which is also illustrated in the “Group Summary” panel. Therefore, we were only able to simulate constant covariate effects but not spatially varying covariate effects or residual spatial effects with SRTsim.

Figure R1. A screenshot of SRTsim’s Shiny application.

2. I think Figure R2 is informative and clear. I suggest put this plot in the main manuscript, maybe in the introduction or in the early part of the results section.

Response:

We thank the reviewer for this suggestion. We have moved this overview figure to the first part of the Results section. It is now Figure 2 in the revised manuscript.

3. In Figure 5B and 5C, what does "spatial-associated" mean? Do they correspond to "spatially varying covariate effect" or "residual spatial effect"?

Response:

We have clarified the explanations of these genes in the main text on page 13. The “layer-associated” genes refer to genes whose expression was layer-associated but did not have residual spatial effects; the “spatial-associated” genes refer to genes whose

expression had residual spatial effects but was not layer-associated. In Figure 6 (previously Figure 5), we use these abbreviations for simplicity.

4. For reproducibility consideration, the authors should submit all the codes to generate the analysis results mentioned in the paper (including numbers, figures, tables, etc.) to a public repository, such as GitHub. Also, the authors should provide necessary annotations to help readers to reproduce the results.

Response:

We thank the reviewer for this suggestion. We have submitted the source code for generating the analysis results (including annotated figure or table numbers) to the zenodo repository (<https://zenodo.org/records/10946411>), and have included the link and DOI in the revised manuscript.